# Identification and Profiling of a Novel *Bombyx mori latent virus* Variant Acutely Infecting *Helicoverpa armigera* and *Trichoplusia ni*

**DOI:** 10.3390/v15051183

**Published:** 2023-05-17

**Authors:** Thomas-Wolf Verdonckt, Anton Bilsen, Filip Van Nieuwerburgh, Loes De Troij, Dulce Santos, Jozef Vanden Broeck

**Affiliations:** 1Molecular Developmental Physiology and Signal Transduction Research Group, Animal Physiology and Neurobiology Division, Department of Biology, KU Leuven, Naamsestraat 59 Box 2465, 3000 Leuven, Belgium; anton.bilsen@kuleuven.be (A.B.); dulce.cordeirodossantos@kuleuven.be (D.S.); jozef.vandenbroeck@kuleuven.be (J.V.B.); 2Laboratory of Pharmaceutical Biotechnology, Ghent University, Ottergemsesteenweg 460, 9000 Ghent, Belgium; filip.vannieuwerburgh@ugent.be

**Keywords:** RNAi, viral piRNA, antiviral immunity, *Bm*N4, High Five, *Maculavirus*

## Abstract

Insect cell expression systems are increasingly being used in the medical industry to develop vaccines against diseases such as COVID-19. However, viral infections are common in these systems, making it necessary to thoroughly characterize the viruses present. One such virus is *Bombyx mori latent virus* (BmLV), which is known to be specific to *Bombyx mori* and to have low pathogenicity. However, there has been little research on the tropism and virulence of BmLV. In this study, we examined the genomic diversity of BmLV and identified a variant that persistently infects *Trichoplusia ni*-derived High Five cells. We also assessed the pathogenicity of this variant and its effects on host responses using both in vivo and in vitro systems. Our results showed that this BmLV variant causes acute infections with strong cytopathic effects in both systems. Furthermore, we characterized the RNAi-based immune response in the *T. ni* cell line and in *Helicoverpa armigera* animals by assessing the regulation of RNAi-related genes and profiling the generated viral small RNAs. Overall, our findings shed light on the prevalence and infectious properties of BmLV. We also discuss the potential impact of virus genomic diversity on experimental outcomes, which can help interpret past and future research results.

## 1. Introduction

Members of the species *Bombyx mori latent virus* (BmLV, formerly also known as *B. mori Macula-like virus*) have a positive, single-stranded RNA genome (Class IV Baltimore). BmLV was first discovered in *Bm*N4 cells, which it persistently infects [1]. It has a genome of 6.5 kb encoding a replicase polyprotein (Rep), a coat protein (CP), and a protein termed p15, which is essential to establish infection although its exact function is unknown [2]. While this species is only known to infect insect cells, it is most closely related to the plant-infecting family of *Tymoviridae* [3], genus *Maculavirus* [4]. Since its discovery, other maculaviruses have been found to infect insects, including bees and mosquitoes [5,6,7,8]. BmLV have been found to persistently infect multiple *B. mori-*derived cell lines, including *Bm*N, NIAS-*Bm*-oyanagi1, SES-BoMo-J125K5, and NIAS-BoMo-Cam1 [1,2], as well as the *Trichoplusia ni*-derived High Five™ (Hi5) cell line [9]. Furthermore, while a BmLV isolate was found capable of only penetrating but not replicating in multiple lepidopteran cell lines (*Sf*9, RP-*Hz*GUT-AW1), it could replicate in some others (*Cf*203, *Sl*2) [9,10,11]. Interestingly, some conflicting results are available in the literature regarding the BmLV tropism. Specifically, some studies reported that BmLV infected Hi5 cells, whereas others reported that the species did not infect this cell line [9,12,13].

Although BmLV is mainly known to maintain asymptomatic, persistent infections, various cytopathic effects might occur when naïve cell lines are treated with the virus. Specifically, increased doubling time, syncytia formation, granularity, and clumping were observed in the *Bm*VF cell line, up to at least six months after infection [14]. Nevertheless, the virus was never found to cause cell mortality. BmLV has not yet been discovered in vivo [11]. However, the virus seemed capable of penetrating *B. mori* midgut cells and expressing its p15 gene [11]. In addition, the expression of viral genes and the presence of viral small (s)RNAs has been observed in *B. mori* ovaries upon BmLV injection into the hemocoel, although at relatively low levels when compared to *Bm*N4 cells [15].

The details of the BmLV replication cycle remain yet unclear. In general, once viral particles penetrate the cell, the virion is uncoated and the viral RNA is released into the cytoplasm. Here, the viral RNA is translated to produce all proteins necessary for replication and transcription. Replication is initiated within the viral factories, with the synthesis of a complementary genomic RNA through the virus-encoded RNA-dependent RNA polymerase (RdRp). This complementary strand then serves as template to produce new genomic RNAs. A 1.25 kb sized subgenomic RNA, which encodes the coat and p15 proteins and corresponds to the 3′-terminus of the viral genome [1], is produced concurrently and in higher amounts than the genomic RNA [2].

During the replication cycle, double-stranded (ds)RNA molecules are formed as viral replication intermediates. These RNA duplexes trigger the RNA interference (RNAi) mechanism, an important insect antiviral response. Specifically, viral dsRNA is recognized within the cytoplasm by the enzyme Dicer-2, which dices it into small-interfering RNAs (siRNAs, 18–24 bp). These siRNAs are then incorporated into an RNA-induced silencing complex (RISC) to direct target-RNA recognition by Watson–Crick base pairing. At this point, the RISC protein Argonaute-2 mediates the degradation of the complementary viral RNA, thus combating the viral infection. This mechanism is considered the most efficient and broadly acting antiviral response in insects [16,17]. In the case of BmLV, siRNAs were identified in the lepidopteran cell lines Hi5, *Bm*N, and *Bm*N4 [2,12,13,18]. Furthermore, Dicer-2 and Argonaute-2 were shown to play a role in the response against this virus in Hi5 cells [12].

In addition to siRNAs, P-element-induced wimpy testis (PIWI)-interacting RNAs (piRNAs) have also been reported to hold antiviral functions in some cases. The piRNA pathway is well-known for its role in the control of transposable elements (also called transposons) in the germline. piRNAs differ from siRNAs in two fundamental aspects: First, they associate with PIWI proteins, which belong to the PIWI clade of the superfamily of Argonaute proteins (in contrast, Argonaute-2 belongs to the AGO clade). Second, their biogenesis is Dicer-independent. The biogenesis and mechanisms of action of piRNAs vary among species, cell type, and even the specific functional purposes. In the model organism *B. mori*, the piRNA pathway is initiated when antisense single-stranded (ss)RNAs are transcribed from genomic piRNA clusters, which are highly rich in transposon remnants. These ssRNA precursors are then processed by multiple factors into mature piRNAs (27–29 nt)—this is the primary piRNA biogenesis pathway. Subsequently, these antisense-piRNAs bind to the PIWI protein silkworm PIWI (SIWI) and find the target transposon transcript, slicing it and marking it for processing into sense-piRNAs. These molecules then bind to a second PIWI protein, named Argonaute-3, to slice antisense piRNA precursors and mark them for processing into new piRNAs. Subsequently, these are loaded into another SIWI protein, and so forth. This process is designated the ping-pong amplification cycle, or secondary piRNA biogenesis, and allows that only piRNAs from active transposons are amplified. Of note, piRNAs deriving from the ping-pong amplification cycle can be promptly identified due to a 9 nt distance bias between the 5′ ends of sense and antisense piRNAs. Whereas primary and secondary piRNA biogenesis pathways are known to be active in the germline, the situation in the soma is not so clear. In the soma of *D. melanogaster* only the primary biogenesis is active; this seems to be different for many other species, including *B. mori*, in which both the primary and secondary pathways are active in somatic tissues. Another indicator that the piRNA pathway holds species-dependent characteristics is the fact that the number of PIWI proteins is highly variable in insects. Extreme cases are exemplified by *B. mori* (Lepidoptera) and *Acyrthosiphon pisum* (Homoptera), which encode two and ten PIWI proteins, respectively. Even within the order of Diptera, pronounced differences are observed, with *D. melanogaster* and *Aedes aegypti* expressing three and eight PIWIs, respectively. Interestingly, an antiviral role of the piRNA pathway in mosquitoes is well established. [19,20,21]. Although this seems to not be the general case in insects, a role of PIWI proteins in antiviral immunity was demonstrated in *B. mori*-derived *Bm*N4 cells and in *T. ni*-derived Hi5 cells [2,18,22].

In this study we identified a BmLV variant persistently infecting Hi5 cells. Next, we investigated its infection properties in two noctuid pest species. The Noctuidae family was selected since it is known for its pest species, which have a large impact on the global economy and are prone to develop resistance towards insecticides. Specifically, we carried out in vitro assays on the BmLV-free fat-body-derived IOZCAS-*Ha*-I cell line [23] and in vivo experiments in the cotton bollworm, *Helicoverpa armigera*. Besides characterizing the viral infection profile in these systems, we observed co-localization of mitochondria and viral replication. In addition, we investigated the regulation of key effectors of the siRNA and piRNA pathways during BmLV infection, namely of Dicer-2, Argonaute-2, and SIWI, as well as characterized their viral sRNA profiles.

## 2. Materials and Methods

### 2.1. Bioinformatics

#### 2.1.1. Virus Discovery

Publicly available sRNA sequencing libraries were imported from the sequence read archive (SRA) [24] onto a local Galaxy platform version 20.09 [25] with the “Faster Download and Extract Reads in FASTQ” tool version 2.11.0. The read quality was assessed and adapters identified using the “FastQC” tool version 0.72 [26]. Adapters were trimmed and reads filtered by size with the “Cutadapt” tool version 1.16.8 [27]. Curated sRNA libraries were then processed through an optimized vdSAR pipeline [28], using an automated workflow. First, contigs were assembled with the Velveth tool version 1.2.10.3 and Velvetg tool version 1.2.10.2 [29] setting a hash length of 17, minimum contig length of 50, and automatic coverage cutoff value. The generated contig list was used as a query for the “NCBI BLAST+ blastn” tool version 2.10.1 [30], using the megablast algorithm on the NCBI “Refseq viruses representative genomes” database downloaded on 21 June 2021, with an expectation cutoff value of 0.0001. The hits were grouped by subject title and their bitscores summed using the “Group” tool version 2.1.4. The sum of the bitscores was used as an initial indicator of viral prevalence.

Transcriptomic sequencing libraries were also used for virus discovery. To do this, Illumina paired-end reads were again imported from the sequence read archive onto the local Galaxy platform with the “Faster Download and Extract Reads in FASTQ” tool. The transcripts were then assembled de novo (no guide genome) with the “Trinity” tool Galaxy version 2.9.1 [31], with a minimal contig length of 200 nt and in silico normalization of reads. The contig list was processed as for the sRNA pipeline.

#### 2.1.2. Genome Assembly

Sequencing libraries with a high BmLV prevalence were used to assemble novel viral genomes. The BmLV genomic sequence (GenBank AB624361.1) was used as the subject for a “NCBI BLAST+ blastn” search using the contig list generated using the Velvetg tool in the virus-discovery workflow. Hits with an expectation cutoff value of 0.001 were filtered from the contig list with the “Filter sequences by ID” tool version 0.2.7. Next, using the GenBank AB624361.1 sequence as the guide sequence, the filtered contig list, and the blastn output, a scaffold was assembled through the “blast_to_scaffold” tool version 1.1.0. The resulting scaffold was then employed as the reference genome to align reads from the curated sRNA library using the “sR_bowtie” tool Galaxy version 2.2.0 [32], allowing two mismatches. The alignments were manually inspected to detect possible SNPs and the assembled genome curated using UGene version 1.29.0 [33]. This was achieved by importing the alignment file to UGene and setting the newly assembled scaffold as the reference sequence. SNPs were visualized by highlighting differences in the aligned reads towards the reference. The scaffold was then manually curated to remove the SNPs.

#### 2.1.3. vsRNA Analysis and piRNA Fingerprinting

To assay the coverage of sRNAs mapping onto the viral genomes and their size, the curated sRNA libraries were aligned to the newly assembled BmLV genomes with the “sR_bowtie” tool, allowing one mismatch. The size distribution and genomic mapping location of aligned reads were obtained by setting the bam alignment files as inputs for the “Generate readmap and histograms from alignment files” tool Galaxy version 1.2.7.4. piRNA base frequency matrices and read distance plots were generated in R via the viRome package [34], using the bam alignment files as input. 

#### 2.1.4. Phylogenetic Analysis

To determine the phylogenetic relationship between BmLV and other *Tymoviridae*, the organization and open reading frames (ORFs) of assembled BmLV genomes were assessed using SnapGene version 6.1.1 (Insightful Science, San Diego, CA, USA). The replicase polypeptide amino acid sequences of the newly identified (and reference) BmLV variants and of published *Tymoviridae* (the potato virus x was included as the outgroup) were aligned using the “Muscle” tool Galaxy version 1.0.0 [35]. The alignment was visualized with MView version 1.63 [36] coloring the amino acids according to their identity. A maximum likelihood phylogenetic tree was generated using the “IQ-tree” tool Galaxy version 2.1.2 + galaxy2 [37], determining the best-fit viral amino acid substitution model with ModelFinder [38], and performing 1000 ultrafast bootstraps [39]. The resulting tree was uploaded to iTOL [40] for rendering. A percentage identity matrix was generated for BmLV replicase polypeptide amino acid sequences with Clustal Omega (ClustalO) [41]. Next, the phylogenetic relationship between the newly identified (and reference) BmLV variants was determined through the replicase polypeptide nucleic acid sequences. Briefly, the ORFs were aligned with the “Muscle” tool and a maximum likelihood phylogenetic tree was generated using the “IQ-tree” tool as before, except the sequence type parameter was set to DNA. All species names and sequence identifiers used during the phylogenetic analysis are provided in Appendix A.

### 2.2. Isolation of BmLV from Hi5 Cells

Hi5 cells (BTI-Tn-5B1-4) persistently infected with BmLV [9] were cultured as described previously [18]. The cells were loosened by scraping, and the viability and concentration were determined by mixing 10 µL cell suspension with 10 µL 0.4% trypan blue (Merck, Darmstadt, Germany) and loading the sample in a TC20™ automated cell counter (Bio-Rad, Hercules, CA, USA).

A total of 2.5 × 10^8^ cells were pelleted by spinning for 10 min at 500× *g* and 4 °C. The supernatant was collected, and the pellet lysed with 4 freeze–thaw cycles in liquid nitrogen. The cell pellet was then resuspended in 15 mL phosphate-buffered saline (PBS, Merck) and spun for 15 min at 500× *g* and 4 °C. The supernatant was collected and spun at 33,100 rpm (ca. 80,000× *g*) for 1 h at 4 °C in an SW 65-Ti rotor (Beckman, Brea, CA, USA). A total of 7.5 mL of the resulting supernatant was concentrated to 700 µL with a Centriprep 30 K centrifugal filter unit (Merck) following the manufacturer’s instructions, then filtered through a 0.22 µm Durapore^®^ Ultrafree^®^ centrifugal filter (Merck).

### 2.3. Infection of IOZCAS-Ha-I by BmLV

IOZCAS-*Ha*-I cells [23] were grown in monolayers at 27.5 °C in TNM-FH insect medium (Merck) supplemented with 10% fetal bovine serum (Merck), 100 U/mL penicillin (Thermo Fisher Scientific, Waltham, MA, USA), 100 mg/mL streptomycin (Thermo Fisher Scientific) and 0.25 µg/mL amphotericin B (Merck). This mixture will henceforth be referred to as “complete medium”. The cells were passaged 1/5 at 90% confluency. 

To perform the infection, 90% confluent cells were loosened by scraping. The cell viability and concentration were determined as before. A total of 6 × 10^6^ cells were pelleted by spinning at 500× *g* for 5 min at 4 °C and resuspended in 6 mL PBS. The cells were split in two equal volumes and pelleted as before. The two cell pellets were then resuspended in 550 µL basal TNM-FH medium each. The two samples were treated as follows: The control cells were treated by adding 50 µL PBS. The experimental cells were treated by adding 30 µL of the virus concentrate, prepared as described in Section 2.2, and 20 µL PBS.

The samples were incubated for 2 h at room temperature (RT) with shaking. The cells were then pelleted as before. The supernatant was removed, and the cells were resuspended in 2 mL PBS, pelleted once more as before, and finally resuspended in 30 mL complete TNM-FH medium. The cell suspension was transferred to 24-well plates at 0.5 mL per well and the plates were incubated at 27.5 °C. A total of 1 mL of the remaining cell suspensions was collected as the time 0 (T0) samples. The T0 cells were spun at 500× *g* for 15 min at 4 °C and the supernatants were removed. The T0 cell pellets were stored at −80 °C. 

Cells were collected at regular intervals by loosening them through pipetting and the concentration and viability were assayed with a TC20™ automated cell counter as described before. The cells were then spun, supernatants removed, and pellets stored as described for the T0 samples.

### 2.4. Helicoverpa armigera Rearing, Infection with BmLV, and Dissection

#### 2.4.1. *H. armigera* Rearing

*Helicoverpa armigera* eggs were acquired from Andermatt Nederland BV. The insect colony was kept for 3 generations, taking care not to couple siblings to prevent inbreeding. The animals were reared at 24 °C with a 14:10 light/dark cycle. The larvae were kept individually in wells of a 6-well plate containing artificial diet (Appendix A).

#### 2.4.2. *H. armigera* Infection with BmLV

Early fourth-instar larvae were starved overnight. The following day, the animals were chilled for 2 h at 4 °C. Next, larvae weighing under 60 mg were briefly put on ice to anesthetize, washed with 70% *v*/*v* EtOH in water, dried, and injected with 3 µL of extract, either from control or from experimentally treated cells (see next paragraph). Injections were carried out laterally on the penultimate abdominal segment with PB600-1 repeating dispensers and 700-series microliter syringes (Hamilton). The animals were then placed back onto the artificial diet and reared as described before. 

The larvae were injected with extracts of control or BmLV-infected IOZCAS-*Ha*-I cells. For the BmLV-infected condition, 7 × 10^6^ IOZCAS-*Ha*-I cells were pelleted by spinning at 500× *g* for 5 min at 4 °C. The supernatant was removed, and the pellet resuspended in 5 mL PBS. The cells were once more pelleted as before and resuspended in 1.283 mL basal TNM-FH medium. The cells were treated with 70 µL of the viral concentrate, prepared as described in Section 2.2, and 47 µL PBS. The cells were incubated for 2 h at RT with gentle shaking on a rocking platform shaker (VWR). The cell suspension was diluted to 5 mL with PBS and the cells pelleted as before. The cell pellet was then washed with 5 mL PBS, pelleted as before, and finally resuspended in 50 mL of complete TNM-FH medium. The cell suspension was distributed equally between two 150 cm^2^ culture flasks and incubated for 5 days at 37.5 °C. Next, the cells were scraped loose, the two volumes combined, and the cell count determined as described in Section 2.2. An equal number of uninfected cells was collected, and cell lysates were prepared in parallel from the infected and uninfected cells as follows: The cells were pelleted by spinning at 1000× *g* for 10 min at 4 °C. The supernatants were removed (supernatants from infected cells were stored at −80 °C), and the cells resuspended in 800 µL PBS. The samples were freeze–thawed four times with dry ice. The cells were spun at 10,000× *g* for 15 min and 4 °C and the supernatants filtered with 0.22 µm Durapore^®^ Ultrafree^®^ centrifugal filters (Merck) following the manufacturer’s instructions. The cell lysates were aliquoted and stored at −80 °C until further use.

#### 2.4.3. *H. armigera* Dissection

Larvae were dissected on day 6 after injection. The larvae were chilled for 2 h at 4 °C, then placed on ice. Before dissecting, the larvae were briefly surface decontaminated by immersion in 70% *v/v* EtOH in water. The animals were then submerged in ice-cold PBS and dissected. The tissues collected were the testes; the ganglia and central nerve cord (henceforth referred to as the central nervous system, CNS); a sample of the fat body; and a section of the midgut spanning from the stomodeal valve until the point of attachment of the Malpighian tubules, with the gut contents and peritrophic membrane removed. The dissected tissues were washed with PBS, flash-frozen in liquid nitrogen, and stored at −80 °C.

### 2.5. Total RNA Extraction

#### 2.5.1. From Cells and Tissues

Total RNA was extracted from the frozen IOZCAS-*Ha*-I cell pellets or *H. armigera* tissues with the miRNeasy Mini Kit (Qiagen, Venlo, The Netherlands), including the DNase digestion step, following the manufacturer’s instructions. The tissue samples were homogenized by shredding in MagNA Lyser Green Bead Tubes (Roche, Basel, Switzerland) with a MagNA Lyser instrument (30 s, 6500 rpm, Roche). A fixed volume of 300 µL (cells) or 350 µL (tissues) was sampled from the aqueous phase, and the purified RNA was eluted in 25 µL RNase-free water.

#### 2.5.2. From Whole Insects

For each timepoint and condition, collected insects were selected at random and pooled in four groups of three. The frozen cadavers were crushed to a fine powder using a liquid-nitrogen-cooled mortar and pestle. A 50 mg sample of powdered tissue was disrupted in 1 mL QIAzol lysis reagent (Qiagen) by shredding (30 s, 6500 rpm) in MagNA Lyser Green Bead Tubes (Roche) with a MagNA Lyser instrument (Roche). The total RNA was extracted using the RNeasy Lipid Tissue Mini Kit (Qiagen) following the manufacturer’s instructions. The DNase digestion step was included to prevent DNA contamination. The purified RNA was eluted in 50 µL RNase-free water. 

For sRNA sequencing, total RNA was extracted from 50 mg of selected samples using the miRNeasy Mini Kit (Qiagen), including the DNase digestion step, following the manufacturer’s instructions. 

### 2.6. cDNA Synthesis

The quality and concentration of extracted RNAs was assayed on a NanoPhotometer^®^ N60 (Implen, Munich, Germany) using a conversion factor of 40 µg/mL/A_260_. Samples with A_260_/A_230_ values under 2, indicating impurities, were further purified through ethanol precipitation. Briefly, the RNA samples were diluted to 100 µL using RNase-free water. A total of 10 µL of 3 M sodium acetate was added to the samples and mixed. Then, 275 µL of 100% EtOH was added, and the samples mixed. The mixtures were incubated for two hours at −20 °C, after which the RNA was pelleted via centrifugation at 12,000× *g* for 15 min at 4 °C. The supernatants were discarded, and the RNA pellets washed with 500 µL of 70% EtOH *v/v* in RNase-free water. The samples were once more spun at 12,000× *g* for 2 min at 4 °C and the supernatants removed. The RNA pellets were allowed to dry and then redissolved in RNase-free water.

Up to 400 ng of RNA was used for a 10 µL final volume cDNA synthesis using the PrimeScript First Strand cDNA Synthesis Kit (TaKaRa, Shiga, Japan) following the manufacturer’s instructions. The cDNA was then diluted by adding 245 µL Milli-Q^®^ water (Merck) (henceforth abbreviated to “MQ”).

### 2.7. Quantitative Real-Time PCR

The expression of viral or host genes was determined on a QuantStudio™ 3 real-time PCR system (Thermo Fisher Scientific). The reaction mixtures and parameters, primer efficiency determination, and reference gene selection were carried out as previously described [42,43,44]. To compare the BmLV titers between *T. ni* and *H. armigera* samples, primers were designed with perfect complementarity towards housekeeping genes of both species and identical amplicon size. All primers employed are given in Appendix A.

### 2.8. IOZCAS-Ha-I Species Identification and BmLV Genome Sequencing

IOZCAS-*Ha*-I cells were described as being *H. armigera*-derived [23]. The recurring failure of PCR reactions using cDNA obtained from IOZCAS-*Ha*-I cells when using primers specific for *H. armigera,* raised doubts on the species identity of the cell line. Notably, amplifications were successful for primer pairs with high similarity to *T. ni* (data not shown). To confirm the species identity, primers were designed against the *argonaute-2* (*ago2*) gene of *T. ni* [12] and the resulting amplicon was cloned for sequencing. Likewise, to sequence the genome of the in-house BmLV variant, primers were designed against ~1000 nt sized overlapping constructs spanning the entire viral genome. The primer sequences are given in Appendix A. Both cloning processes were carried out on cDNA obtained from sample 1 of day 3 BmLV-infected IOZCAS-*Ha*-I cells. 

The *ago2* construct was amplified in four 25 µL PCR reactions using the Q5^®^ High-Fidelity DNA Polymerase (New England BioLabs, Ipswich, MA, USA) following the manufacturer’s instructions. Each reaction was carried out with a different annealing temperature, namely 53, 55, 57, and 59 °C. The PCRs consisted of an initial denaturation at 95 °C for 1 min, followed by 30 cycles of 95 °C for 1 min, annealing temperature for 2 min, 72 °C for 1 min 30 s, and final elongation at 72 °C for 5 min. 

The BmLV constructs were amplified in 25 µL PCR reactions using the REDTaq^®^ ReadyMix™ DNA polymerase (Merck) following the manufacturer’s instructions. Annealing temperatures were as follows: constructs 1, 3, and 4 at 60 °C; construct 2 at 59 °C; construct 5 at 58 °C; construct 6 at 62 °C; construct 7 at 57 °C; and constructs 8 and 9 at 61 °C. The PCRs consisted of an initial denaturation at 94 °C for 30 s, followed by 30 cycles of 94 °C for 30 s, annealing temperature for 30 s, 72 °C for 2 min, and final elongation at 72 °C for 5 min.

The PCR products were loaded in a 0.5% agarose gel stained with GelRed (Biotium, Fremont, CA, USA) in TAE buffer, using an O’RangeRuler 200 bp DNA ladder (Thermo Fisher Scientific) for size estimation. The gel was run at 120 V for 40 to 60 min. The gel was then visualized on a ProXima 2000 series platform (Isogen, De Meern, The Netherlands), and the bands of expected size excised under UV light. The DNA was purified from the agarose gel slices using the GenElute™ Gel Extraction Kit (Merck) following the manufacturer’s instructions. 

To add 3′ deoxyadenosine overhangs to the *ago2* construct, 6.5 µL gel extract was combined with 6.5 µL REDTaq^®^ ReadyMix™ (Merck) and incubated for 5 min at 72 °C. 

Next, 4 µL of the amplified constructs was combined with 1 µL pCR^®^ 4-TOPO^®^ vector and 1 µL salt solution from the TOPO TA Cloning^®^ Kit (Thermo Fisher Scientific) and the samples were incubated at room temperature for 15 min. One Shot™ Top10 chemically competent cells (Thermo Fisher Scientific) were thawed on ice and 5 µL of the constructed 4-TOPO^®^ vectors added. The whole sample was mixed gently by pipetting and incubated on ice for 30 min. The bacterial cells were heat-shock transformed by placing the tube with bacteria in a 42 °C water bath for 40 s. The cells were then quickly cooled on ice. A total of 250 µL SOC medium (Thermo Fisher Scientific) was added to the cells and they were incubated at 37 °C with shaking for 1.5 h (*ago2* construct) or 3 h (BmLV constructs). Then, 150 µL of cell suspension was distributed over LB agar plates containing 50 µg/mL ampicillin and incubated at 37 °C overnight. The following day, individual colonies were carefully scraped from the plate and resuspended in 20 µL MQ water. 

Bacterial clones were verified through colony PCR. For each colony, 2 µL of the bacterial suspension was combined with 2 µL of 10 µM forward primer, 2 µL 10 µM reverse primer (sequences available in Appendix A), and 6.5 µL REDTaq^®^ ReadyMix™ (Merck). The PCR reaction was run for 25 cycles as before, with an annealing temperature of 55 °C and elongation time of 45 s. Finally, an elongation step of 5 min at 72 °C was carried out. The PCR products were loaded on a 1% agarose gel stained with GelRed (Biotium) in TAE buffer, and run and visualized as before. For colonies that gave a band of expected size, the bacterial suspension in MQ was added to 5 mL LB medium with 50 µg/mL ampicillin and incubated at 37 °C with shaking overnight. The plasmids were extracted from 2 mL of the cultured cells using the GenElute™ HP Plasmid Miniprep Kit (Merck) following the manufacturer’s instructions. The cloned inserts were sequenced in triplicate using the cloning primers through Sanger sequencing (LGC Genomics, Berlin, Germany). 

The *ago2* sequencing results were aligned, and a consensus generated with UGene. The consensus was set as query for a megablast search towards the NCBI nucleotide collection (nr/nt) updated 25 September 2022 or the NCBI transcriptome shotgun assembly (TSA) sequences collection updated 25 September 2022. 

The BmLV sequencing results were similarly assembled, and a consensus generated with UGene.

### 2.9. Confocal Immunofluorescence Microscopy

Glass coverslips (0.17 mm thick and 20 × 20 mm wide) were treated with 70% EtOH, allowed to dry, and placed in 6-well cell culture plates. A total of 2 mL of cell suspension in complete medium was added to the wells and the cells allowed to attach for three hours. IOZCAS-*Ha*-I cells were infected with BmLV by adding 1 mL of the supernatant collected from infected cell medium described in Section 2.4.2. The cells were incubated at 27.5 °C overnight. The following day, the old medium was removed and replaced with 2 mL of prewarmed complete medium supplemented with 200 nM of MitoTracker™ Deep Red FM dye (MTDR) (Thermo Fisher Scientific). The cells were stained for 1 h at 27.5 °C after which the cell medium was removed, and the coverslips washed with 1 mL PBS. The following steps were carried out at room temperature unless stated otherwise. The coverslips were then transferred to 35 mm petri dishes and again washed with 1 mL PBS. The cells were fixed with 250 µL of 4% paraformaldehyde (PFA) (Merck) for 10 min, then washed twice with 1 mL PBS. The cells were then permeabilized with 1 mL 0.2% Triton-X 100 (Merck) in PBS for 10 min and again washed thrice with 1 mL PBS, incubating them each time for 5 min. Next, the cells were treated with 1 mL of 2 µg/mL (1/500 dilution) anti-dsRNA monoclonal antibodies J2 (Jena Bioscience) in PBS. The samples were incubated overnight at 4 °C with gentle shaking. The following day, the cells were washed thrice with 1 mL PBS, incubating them each time for 5 min. The cells were then stained for 45 min with 2 droplets of Alexa Fluor™ 488 goat anti-mouse ReadyProbes™ secondary antibodies in 1 mL PBS with 0.2% bovine serum albumin (BSA) (Merck); this and the following steps were carried out in low-light conditions. Next, the cells were stained for 3 min with 150 µL of 1 µg/mL 4′,6-diamidino-2-phenylindole (DAPI) (Thermo Fisher Scientific) and 1.5 µg/mL phalloidin–tetramethylrhodamine B isothiocyanate (Phalloidin–TRITC) (Merck) in PBS. The cells were then washed thrice with 1 mL of 0.05% Triton-X 100 in PBS, twice with 1 mL of distilled water for 1 min, and finally mounted on a microscope slide with 5 µL Mowiol 4–88 mounting medium (Merck). The slides were allowed to harden at room temperature and shielded from light for several hours before being imaged on a FluoView FV1000D-IX81 laser confocal scanning microscope (Olympus, Shinjuku, Tokyo, Japan).

### 2.10. Statistical Analysis

All statistical analyses were carried out with GraphPad Prism software version 8. Differences in the growth curves of mock- and BmLV-treated IOZCAS-*Ha*-I cells were assessed by comparing the best-fit values of logistic growth models computed through least squares regression, setting a fixed Y_0_ value of 0.7. Differences in the amount of RNA in wells of mock- and BmLV-treated IOZCAS-*Ha*-I cells were likewise assessed by comparing the best-fit values of logistic growth models computed through least squares regression; however, without setting initial values. Differences in the RNA content per well, cell concentration, or cell viability on a day-by-day basis were assessed by performing a two-way ANOVA with post hoc Sidak’s multiple comparisons test.

For each IOZCAS-*Ha*-I gene, the transcript levels were normalized to the average of the T0 timepoint using the ddCt method. Next, variations in the transcript levels over time were assessed with an ordinary one-way ANOVA. A post hoc Dunnett’s multiple comparison test was carried out against the 4 h timepoint.

Differences in weight between mock- and virus-infected larvae were assessed on a day-by-day basis with multiple t-tests. The mortality and timing of developmental events of mock- and virus-infected animals were compared with Log-rank (Mantel–Cox) and Gehan–Breslow–Wilcoxon tests.

The expression of RNAi-pathway-related genes in *H. armigera* animals following mock- or viral-infection was monitored by normalizing the gene transcript levels towards the mean of the mock-infected samples from the same day using the ddCt method. Differences in transcript levels between treatments were then compared with multiple Welch’s t-tests, as the samples were not expected to have equal variance.

### 2.11. Next Generation Sequencing

sRNA library preparation and sequencing were performed at the NXTGNT facility (Universiteit Gent). First, the integrity of the total RNA samples was assayed on a 2100 Bioanalyzer instrument (Agilent Technologies, Santa Clara, CA, USA) on an RNA nano chip. The concentration of the samples was then measured with the Quant-it Ribogreen RNA assay kit (Thermo Fisher Scientific). Libraries were prepared using 140 ng total RNA and the Qiaseq miRNA library kit (Qiagen) following the manufacturer’s instructions with 16 amplification cycles. The quality of the amplified libraries was controlled on a 2100 Bioanalyzer instrument using a DNA high-sensitivity chip. The PCR product was then purified through electrophoresis on a native 8% polyacrylamide gel, and the product was once more quality controlled on a DNA high-sensitivity chip. The resulting libraries were quantified using qPCR according to Illumina recommendations, and equimolarly pooled. Sequencing was performed on a NextSeq500 system (SR75, high output) (Illumina, San Diego, CA, USA).

## 3. Results

### 3.1. Prevalence and Phylogeny of BmLV in Lepidopteran Cell Lines and Insects

We started our study by investigating the prevalence and diversity of BmLV in Lepidoptera. For this, we analyzed sRNA SRA datasets from 37 publicly available BioProjects obtained from insect cell lines or in vivo, representative of 9 species (Appendix A). We used an optimized vdSAR pipeline to perform viral identification on each dataset. Of all analyzed datasets, BmLV was only identified in the ones derived from *Bm*N4 cells or from cells where it was experimentally administered, namely the *Bm*VF cells.

Furthermore, BmLV was detected in all *Bm*N4-derived BioProjects (Appendix A), independent of sampling timing and geographical location. Considering this finding, we questioned the diversity of these BmLV genomes. Therefore, we further investigated this by assembling genomes from select sRNA SRA databases with abundant viral reads. As an additional control, for BioProjects that contained sRNA SRA and TSA databases, the BmLV transcript was identified in the TSA and compared to the genome assembled from the sRNA SRA. The genomes of BmLV infecting Hi5 cells in our lab were also reconstituted. In total, sixteen genomes were assembled, one from each dataset, obtained within five countries (Appendix A). For each sRNA library, the viral abundance was estimated through the ratio of virus-targeting reads to the total reads, as expressed in reads per million (RPM) (Appendix A). In addition, the BmLV genome of one of our stocks of Hi5 cells (FHV-free) was confirmed using cloning and Sanger sequencing (Appendix A). When aligned with the *Bm*N4-derived BmLV reference genome (GenBank AB624361.1), a 95.31% amino acid identity was observed for the replicase polyprotein (Appendix A). 

Afterwards, to gain insight into the evolutionary history of the sixteen assembled BmLV genomes, we performed a phylogenetic analysis. First, the translated amino acid sequences of the replicase polyprotein (Figure 1a green, Appendix A) were used to construct a maximum likelihood phylogenetic tree (Figure 1b). 

The potato virus x (PVX) was used as the outgroup and a selection of known Marafiviruses, Maculaviruses, and Tymoviruses was used for clade determination. The previously described and the here assembled BmLV genomes clustered in one branch (collapsed and indicated by the triangle) and, together with the BeeMLV2, form the *Maculavirus* clade (Figure 1b). Next, the phylogenetic relationship between the BmLV genomes was determined through the replicase polyprotein ORF nucleic acid sequences (Figure 1c). The BmLV variants clustered into distinct clades that mainly correlate to their geographic location and host cell line (Figure 1c).

### 3.2. Prior Characterization of the Lepidopteran IOZCAS-Ha-I Cell Line

The lepidopteran IOZCAS-*Ha*-I cell line was provided to us in April 2018 by Zhang et al. from the Institute of Zoology, Chinese Academy of Sciences [23]. Upon arrival, the cells were imaged with an inverted microscope (Appendix A). Afterwards, the species of origin of these cells was determined by sequencing a 1.5 kb segment of the *argonaute-2 (ago2)* ORF. This gene was selected since it was found to be one of the fastest evolving genes in *Drosophila* [46,47], increasing the sensitivity to discriminate between closely related species. The sequenced *ago2* segment (Appendix A) had 100% sequence identity with predicted *T. ni ago2* transcripts, while being only 76% identical to the expected *H. armigera* transcript. No other species match was found using a BLAST towards the NCBI nr or TSA databases. Afterwards, the presence of viral infections in this cell line was assessed. For this, total RNA was extracted from IOZCAS-*Ha*-I cells, and the sRNA fraction was sequenced. Viral prevalence was then assayed as described in Section 2.1.1. The analysis yielded no positive hits, suggesting that the cell line was free from known viruses inducing an RNAi response.

### 3.3. BmLV Causes Acute and Lethal Infection in the Lepidopteran IOZCAS-Ha-I Cell Line 

In this study, a BmLV variant was isolated from the *T. ni*-derived Hi5 cells and its infection dynamics were analyzed in the IOZCAS-*Ha*-I cell line. Upon viral infection, the cell concentration, viability, RNA content, and viral titers were assayed at regular intervals from 4 h until 8 days after the infection. Treated cells displayed a stunted growth (logistic growth comparison of fits, *p* < 0.0001), with significantly lower cell density than mock-treated cells from day 4 (two-way ANOVA, *p* < 0.0001) until the end of observations on day 8 (Figure 2a). This was reflected in the total RNA content of infected wells (logistic growth comparison of fits, *p* < 0.0001) which was significantly lower than in mock-treated wells from day 3 (two-way ANOVA, *p* < 0.0003) until the end of observations on day 8 (*p* < 0.0001, Figure 2b). Infected cells also displayed significantly decreased viability starting on day 3 (two-way ANOVA, *p* < 0.0003) which decreased to a mean value of 35% compared to 90% of mock-infected cells on day 8 after the infection (*p* < 0.0001, Figure 2c). Infected cells displayed visible cytopathic effects, including increased clustering, increased cell size, and presence of detritus (Figure 2d). 

### 3.4. BmLV Replication Co-Localizes with Mitochondria

As part of its replication cycle, the BmLV makes use of an RdRp to synthetize an antigenomic RNA strand, leading to dsRNA intermediates. As long dsRNA is not present in healthy eukaryotic cells, it can serve as a marker for viral infection and replication [48]. By labeling dsRNA, localization of viral replication within the cell was assessed. For this, a colocalization study was performed between the virus-derived dsRNA and mitochondria using confocal immunocytochemistry. IOZCAS-*Ha*-I cells were imaged one day after infection with BmLV as described in Section 2.9. dsRNA was labeled with the J2 monoclonal antibody [49] and visualized with Alexa Fluor™ 488 secondary antibodies, mitochondria were labeled with MTDR, the cell nuclei with DAPI, and actin with phalloidin–TRITC. IOZCAS-*Ha*-I cells were readily stained with MTDR, DAPI, and phalloidin–TRITC (Figure 3a). Cells with a strong J2 signal and cells without a J2 signal were detected (Figure 3b) indicating that the detected dsRNA was BmLV-derived. An overlay of MTDR and the J2 signal shows that the viral intermediates of replication colocalized with mitochondria.

### 3.5. Regulation of siRNAi and piRNAi Machinery during BmLV Infection in IOZCAS-Ha-I Cells

The transcript levels of viral and key RNAi genes were assessed via quantitative real-time PCR (RT-qPCR) in BmLV-treated IOZCAS-*Ha*-I cells. For each timepoint, the RNA from mock- or BmLV-infected cells was extracted and cDNA libraries were prepared. The housekeeping genes *α-tubulin* (*α-tub*) and *ribosomal protein S18* (*RPS18*) were found to be the most stable over time and used as internal reference genes. Transcript levels were normalized towards the T0 timepoint using the ddCt method and for each timepoint the mean values were compared to those of 4 h after infection using one-way ANOVA with post hoc Dunnett’s multiple comparison test. The host genes analyzed were two core genes of the small interfering RNAi (siRNAi) pathway, *argonaute-2* (*ago2*) and *dicer-2* (*dcr2*), and one gene of the PIWI-interacting RNAi (piRNAi) pathway, *siwi*. The siRNAi genes *ago2* and *dcr2* displayed a brief upregulation following infection. For *ago2*, the gene was significantly upregulated on days 1, 2, and 3 after infection (*p* = 0.0030, <0.0001, and <0.0001, respectively; Figure 4a), while for *dcr2* an upregulation was observed on days 2 and 3 (*p* = 0.017 and <0.0001, respectively; Figure 4b). The gene transcript levels then returned to T0 values. Notably, *ago2* was significantly (*p* = 0.019) although slightly (mean ± SD values of 0.759 ± 0.0523) downregulated on day 5 after infection. Surprisingly, the transcript levels of *siwi* were significantly (*p* < 0.0001; Figure 4c) downregulated from day 3 after the infection until the end of measurements on day 8, with the lowest value on day 5 (mean ± SD values of 0.540 ± 0.0318). 

In parallel, viral titers were assayed with primers targeting the RdRp domain of the *replicase polyprotein* (*rep*) gene or the *coat protein* (*cp*) gene (Figure 1a). The former served as an indicator of viral genome titers, including negative-strand intermediates of replication. The latter additionally measured the titers of subgenomic RNAs. In IOZCAS-*Ha*-I cells, the BmLV *rep* titers surpassed those in persistently infected Hi5 cells within 1 day after infection (Figure 4d, red dotted line). The appearance of cytopathic effects (Figure 2a–c) corresponded with the plateauing of viral titers on day 3 (Figure 4d). The viral sequences reached a maximum ± SD (relative to T0) expression value of 2.11 ± 0.14 × 10^4^ for *rep* on day 5, and of 2.31 ± 0.27 × 10^4^ for *cp* on day 4. Notably, on day 4 the Ct ± SD values of the BmLV *cp* gene were measured at 7.50 ± 0.08, compared to 21.83 ± 0.08 for *RPS18* and 16.24 ± 0.08 for *α-tub*. This was close to the Ct values of the *ribosomal 18S* RNA at Ct ≈ 6 (Appendix A).

### 3.6. BmLV Causes Acute Followed by Persistent Infection in H. armigera In Vivo 

The infectivity and pathogenicity of BmLV was assayed in an important noctuid pest species, the cotton bollworm *H. armigera*. In IOZCAS-*Ha*-I cells, the viral titers reached their maximal levels 5 days after infection (Figure 4). For this reason, extracts were prepared from control cells or cells 5 days after BmLV-infection. Next, fourth-instar *H. armigera* larvae weighing under 60 mg were injected with 3 µL of control or infected cell extracts, corresponding to an extract equivalent derived from 3 × 10^4^ cells per animal. The larvae were then monitored daily, noting their weight, developmental stage, and deaths (Figure 5). Compared to mock-infected animals, larvae infected with BmLV displayed significantly reduced weight ± SD on day 3 (mean of 249 ± 107 vs. 202 ± 75 mg) and 4 (mean of 379 ± 173 vs. 329 ± 158 mg) after infection but caught up with mock-infected animals as soon as day 5 (Figure 5a). While BmLV infected animals seemingly displayed higher mortality rates than mock-infected individuals, this did not result in significantly different survival curves (Figure 5b). Infection with BmLV also affected the timing of developmental events. Infected animals displayed significantly delayed molting to the fifth instar and pupal stages. However, emergence to the adult stage was not significantly affected (Figure 5c). 

Animals were collected randomly for further analysis at the following timepoints: immediately after injection (T0); after 4 h; on days 1, 2, 3, and 6 after injection; as 1-day-old pupae; and as 1-day-old adults. For each timepoint, the animals were pooled in four groups of three individuals, and whole-body cDNA libraries were prepared. At 6 days post injection, an additional six male larvae were collected, the testis, fat body, CNS, and midgut were individually collected, and cDNA libraries were prepared (Figure 6a displays a schematic diagram of the collected tissues). The viral titers were determined through RT-qPCR, with primers targeting the BmLV *rep* gene (Figure 6b,d) or the BmLV *cp* gene (Figure 6c), and normalized towards the T0 timepoint. Both targets displayed comparable timelines. Four hours after injection, the relative viral titer means decreased (*rep* = 0.52; *cp* = 0.61) but increased on day 1 (*rep* = 4.59; *cp* = 11.28) and reached a maximal value on day 2 (*rep* = 4.70; *cp* = 13.13). The viral titers then decreased and stabilized from day 6 until adulthood. The viral titers in *H. armigera* were compared to those in Hi5 cells (Figure 6b,c). For this analysis, the normalization was carried out with the housekeeping genes *EF1α* and *heat shock protein 90* (*HSP90*). The results indicated that on days 1 and 2 after infection the BmLV *rep* gene was expressed more in vivo than in the persistently infected cell line (Figure 6b). Notably, the viral titers displayed a large standard deviation from day 1 to 3 after infection. The viral load in individual larvae and tissues also varied greatly (Figure 6d). Two larvae had low (≤0.1) *rep* expression in all tissues, while three displayed increased viral loads compared to the T0 timepoint. Of all tissues, the midgut had consistently low viral loads, whereas the fat body had higher viral loads.

### 3.7. Regulation of siRNAi and piRNAi Machinery during BmLV Infection in H. armigera In Vivo

The RNAi-based immune response of the host was investigated by monitoring the transcript levels of the siRNA-pathway genes *ago2* and *dcr2* (Figure 7a,b), as well as the piRNA-pathway gene *siwi* (Figure 7c), for mock- or BmLV-infected animals, as described in the previous section. The *ago2* gene was upregulated in BmLV-infected animals, with an up to 6.9-fold (mean of 4.9-fold) higher expression compared to mock-infected animals on day 1 after the treatment (Figure 7a). This upregulation was stable throughout the larval stage but was less prominent in the adult stage, with a mean upregulation of 2.0-fold (Figure 7b). The *dcr2* gene was also upregulated in BmLV-infected animals, with an up to 25.0-fold (mean of 12.7-fold) higher expression compared to mock-infected animals on day 2 after treatment. The upregulation was present until the adult stage, with a mean value of 3.9-fold. Although *dcr2* upregulation on days 2 and 6 was not statistically significant, a trend was still observed. In this respect, the large standard deviation in the virus-infected group compared to the control group is of note (Figure 7b). Contrary to the other tested genes, *siwi* was only upregulated on day 2 after infection (Figure 7c, mean of 1.7-fold) and did not differ significantly at any other timepoint.

Of note, we identified *H. armigera siwi* genes in two distinct genomic loci, on chromosomes 12 and 17 (Appendix A). The two predicted SIWI proteins (GenBank XP_049696273.1 and XP_049699728.1) were highly similar, with 98% (ClustalO) AA identity. Notably, the codon changes were concentrated at the N-terminus, a region with low degree of domain conservation in PIWI proteins (Appendix A).

### 3.8. BmLV Infection in IOZCAS-Ha-I Cells and in H. armigera In Vivo Is Associated Mainly with Viral siRNAs

The generation of BmLV sRNAs upon infection was investigated. For this, IOZCAS-*Ha*-I cells and *H. armigera* caterpillars were infected with the BmLV as described in Section 2.3 and Section 2.4.2, respectively. The testes, fat body, CNS, and midgut (Figure 6a) of a single male larva were dissected on day 6 after infection (Figure 8, samples 01, 02, 03, and 04, respectively). Three *H. armigera* animals were collected and pooled on day 2 after infection (Figure 8 sample 05) and on day 1 of the adult stage (Figure 8, sample 06). BmLV- and mock-infected IOZCAS-*Ha*-I cells were collected on day 2 after infection (Figure 8, samples 07 and 08, respectively). Total RNA was extracted from each sample and the sRNA fraction was sequenced. The read size distribution of each sample is provided in Appendix A. BmLV-infected IOZCAS-*Ha*-I cells produced large quantities (>2500 RPM) of sense-oriented 20 nt and ~27 nt sized sRNAs targeting the viral genome. By contrast, mock-infected cells contained a negligible amount of sRNAs mapping onto the viral genome, confirming that this cell line is not BmLV-infected (Figure 8a). vsRNAs produced in IOZCAS-*Ha*-I cells targeted the entire viral genome, with hotspots corresponding to the genomic 5′ end and subgenomic RNAs (Figure 8b). 

As *H. armigera* animals were infected with BmLV by injecting extract from infected IOZCAS-*Ha*-I cells (*T. ni-*derived), sRNAs from this cell line could have persisted in the hemolymph and contaminated the *H. armigera* sRNA pools. To exclude this possibility, the sRNA libraries of all samples were mapped onto a *T. ni* PiggyBac transposon sequence (GenBank: DQ340395.1). While both virus- and mock-infected IOZCAS-*Ha*-I datasets gave RPM values over 5000, the library obtained from whole L4 larvae two days after injection remained under 0.1 (Appendix A). Therefore, we considered read alignments with size distribution RPM values above a threshold of 0.2 to be *H. armigera*-derived. BmLV infected *H. armigera* animals produced 20 nt sized BmLV-vsRNAs, with a weak sense bias (Figure 8a), but extremely limited amounts of piRNA-sized sRNAs. The vsRNA abundance remained relatively stable, with a range of ~300–700 RPM from L4 larvae to the adult stage. Within a single *H. armigera* animal, infected tissues showed more variation in vsRNA abundance, with the midgut having the lowest vsRNA fraction (~20–30 RPM) and the fat body the highest (~4000–6000 RPM) (Figure 8a). Interestingly, while the *H. armigera*-derived vsRNAs displayed a hotspot regarding the 5′ genomics end, they did not display hotspots corresponding to the subgenomic RNAs (Figure 8b).

As seen in Figure 8a, for *H. armigera* samples the piRNA-sized reads were barely or not visible when scaled alongside the siRNA peak. Therefore, in Figure 9a, only the 23–31 nt reads are shown. Notably, in the fat body and CNS samples, sRNAs displayed a strong sense bias and peaks of 27 nt were visible. On the other hand, in the testes both sense and antisense reads were produced with a slight sense bias, and the peak was at a shorter length, namely at 25–26 nt. No clear peak could be seen in sRNAs extracted from the midgut, and the RPM values were close to or under the confidence threshold. In whole body extracts of L4 larvae, a similar pattern was seen as in fat body and CNS tissues, whereas in whole body extracts from adults the 27 nt peak became less prominent and antisense reads more represented, possibly due to the larger gonads in this developmental stage.

### 3.9. piRNA Biogenesis Pathways are Active in IOZCAS-Ha-I Cells and in the Studied H. armigera Tissues

To perform a piRNA fingerprinting analysis, sRNA reads from the IOZCAS-*Ha*-I cell line were mapped onto the transposase gene of a *T. ni* PiggyBac transposon identified in the *T. ni*-derived Hi5 cells (GenBank: DQ340395.1). Antisense sRNAs displayed a bias towards an uracil in the first position, and sense sRNAs displayed a bias towards an adenine in the tenth position. Moreover, a read distance plot showed a clear peak at a distance of 9 nt (Appendix A). Regarding the *H. armigera* 27 nt sRNAs, when mapped onto the *H. armigera* retrotransposon *Ha*RT1 (GenBank: EU016079.1), antisense reads displayed a strong 1 U bias, whereas sense reads displayed a strong 10 A bias (Appendix A). A strong 9 nt distance bias between the 27 nt sense and antisense reads was also observed (Appendix A) [50,51].

## 4. Discussion

### 4.1. The Diversity of BmLV Variants and Its Implications

In this study we have characterized the genome of a BmLV variant persistently infecting the *T. ni* Hi5 cells of our lab. Specifically, the genome sequence was first reconstituted from available sRNA databases, followed by confirmation via cloning and Sanger sequencing. In addition, by using publicly available databases, we reconstituted genomes of BmLV infecting *B. mori Bm*N4 cells, which were used for phylogenetic analysis. We found that the BmLV variant infecting Hi5 cells differed markedly from variants infecting *Bm*N4 cells (Figure 1c), with up to a 5% difference in the amino acid sequence of the replicase polyprotein (Appendix A). This is particularly relevant with regard to the current lack of consensus on whether Hi5 cells are permissive to BmLV infection. BmLV was first reported infecting *T. ni* (Noctuidae) cells by Swevers et al. (2016). However, their finding was met with skepticism as it could not always be reproduced [52]. Hi5 cells infected with BmLV were since then reported in two more studies [12,18]. Besides demonstrating beyond doubt that Hi5 cells could be infected by BmLV, our results could explain conflicting reports on BmLV species tropism. Specifically, distinct BmLV variants could display different abilities to infect distinct species. For example, the *Spodoptera frugiperda* (Noctuidae)-derived *Sf*9 cells were repeatedly reported to not be permissive to replication of BmLV extracted from *Bm*N4 cells [10,11]. By contrast, *Sf*9 cells infected with BmLV extracted from Hi5 cells remained PCR positive up to two weeks after infection [9]. Nevertheless, this remains to be investigated and it would be useful to direct future research at assaying the effect of the BmLV variant diversity on viral tropism.

The exact origin of BmLV remains unknown. In *Bm*N4 cells, where it was first described [1], BmLV was present in every analyzed cell stock (Appendix A), suggesting that infection took place before the dissemination of the cell line. It is unlikely that BmLV was already infecting the larvae from which the cells were obtained, as BmLV has not yet been detected in animals, and seems incapable of replicating in them [1]. As for the Hi5-derived BmLV variant used in this study, it might have originated from an accidental contamination with *Bm*N4-derived virus, as only a few Hi5 stocks were infected with BmLV. In addition, our phylogenetic analysis indicated that this strain did not form a distant outgroup in relation to the *Bm*N4-derived BmLV genomes (Figure 1c). Of note, Iwanaga et al. (2012) found that BmLV was present in multiple *B. mori*-derived cell lines. Sequencing analyses revealed that in these cells BmLV underwent several amino acid changes in the replicase, coat protein, and p15 genes [9,14]. Recently, a new variant was discovered with a 501 nt sized deletion in the RdRp gene [11]. The route of infection however remains unclear. Specifically, a recent BmLV cross-contamination from *Bm*N4 or a previous infection followed by cell–virus co-evolution remain as possibilities.

Besides having an impact on viral tropism, it is interesting to hypothesize that different variants of the BmLV could reflect on the host immune response, for instance by triggering/inhibiting specific sRNA pathways or evading from them. We have previously observed that the size distribution of sRNAs targeting BmLV in *Bm*N4 cells could vary between laboratories and hypothesized that these differences are related to the use of distinct *Bm*N4 stocks [18]. In light of our current findings that different *Bm*N4 cell stocks were infected by different variants of the BmLV (Figure 1b,c), it is not possible to exclude that the observed differences might also have been variant-related. The *Bm*N4 cell line was established over half a century ago [53] and, during this long period, both the cell line and the virus may have evolved, possibly leading to variability in antiviral and/or viral mechanisms. In addition, *Bm*N4 cells were reported to readily differentiate into two distinct cell types [54,55], which could possibly provide different contexts for BmLV evolution.

Another implication of the observed BmLV genome diversity is that it could compromise PCR-based detection strategies. Specifically, nucleotide diversity at the primer-binding sites could result in false negative diagnoses. Thus, other techniques such as those based on protein detection might be useful in a first approach. For instance, Western blot is not as sensitive to strain diversity and can even be used to determine closely related viral species [56]. These ideas are in line with the observations of Iwanaga et al. (2012) that cell lines that were determined to be BmLV-negative through PCR, turned out to be infected when analyzed using Western blot. Deep sequencing techniques can also be useful in a viral detection context, as reported here. However, it is important to note that manual curation of sRNA alignments onto the assembled viral genomes revealed the presence of SNPs and, possibly, of multiple viral variants within single SRAs. While the automated vdSAR protocol applied in this research did not account for diversity at this level, we have performed manual curation. Therefore, while automated vdSAR protocols can be used for diagnosis, single genome determination should rely on whole-sequence cloning or long-read sequencing using large and overlapping constructs as was performed in the present study.

### 4.2. Persistent Viral Infections in Cell Lines and Their Implications

When assessing the presence of BmLV infections in lepidopteran cells, we noticed that most cell lines infected by BmLV also harbored additional viral infections (Figure 1c, Appendix A). This includes the *Bm*VF (*B. mori* virus-free) cells, a cell line that was established with the purpose of studying BmLV infection dynamics in naïve cells [14]. Of note, BmLV is known to be an opportunistic virus that shows increased titers during co-infection with a baculovirus [57], an alphanodavirus [9] (Appendix A), and a dicistrovirus [58]. Thus, in addition to a potential impact of the here reported BmLV-variant diversity (Figure 1c), it is relevant to consider that co-infections could contribute to the disparate reports in the literature regarding BmLV tropism [4,9,12,52]. 

Insect cells are important systems to produce recombinant proteins through the baculovirus expression vector system (BEVS) [59], including for veterinary and medicinal use [60]. Recently, the BEVS gained special attention when it was used to produce a SARS-CoV-2 spike glycoprotein in *Sf*9 cells, as part of a vaccine to prevent the coronavirus disease 2019 (COVID-19) [61]. Considering that BmLV persisted in *Sf*9 cells [9], it is of growing concern that the utilized cell lines are free of unwanted pathogens. BmLV seemed incapable of replicating or maintaining infections in a selection of mammalian cell lines [10]. However, it was capable of entering human lung cells, with the viral genome remaining detectable several weeks after treatment [11]. The *Bm*VF cell line was proposed as an alternative to *Bm*N4 cells for recombinant protein production, as *Bm*VF cells were thought to be virus-free [52]. However, our analysis showed that this cell line was persistently infected with a virus belonging to the *Iflaviridae* family (Figure 1 and Appendix A). This finding supports the need for thorough virome determination in insect cell lines prior to their use for recombinant protein production. VdSAR, as proposed by [28] or as optimized here, or TBLASTN-based improved methods as proposed by [62] can be of use for this purpose.

### 4.3. BmLV Acutely Infects Two Noctuidae Species In Vivo and In Vitro

Our results demonstrated that the IOZCAS-*Ha*-I cell line was a *T. ni*-derived cell line (Section 3.2). We reported that these cells were sensitive to infection by the BmLV, which resulted in cessation of cell growth and high mortality (Figure 2). In fact, the BmLV-induced pathogenesis in IOZCAS-*Ha*-I cells was higher than that described for *B. mori* cells. BmLV infection was asymptomatic in *Bm*N4 cells and, in *Bm*VF cells, it correlated with increased doubling time [14]. In line with this, we observed higher viral titers in IOZCAS-*Ha*-I cells than those reported for *Bm*N4 cells: in *Bm*N4 cells, the BmLV *cp* gene could reach copy numbers close to those of *actin3* [1], while in IOZCAS-*Ha*-I the copy numbers related to those of 18S RNA, one of the most abundantly expressed RNAs in eukaryotic cells (Appendix A). While the mechanisms leading to BmLV-induced mortality in IOZCAS-*Ha*-I cells remain unclear, the high levels of viral genes suggest that cell death may have occurred due to the complete allocation of cell resources to viral production. This is supported by the synchronicity of plateauing viral titers and the appearance of cell mortality (Figure 2 and Figure 4).

In *H. armigera* in vivo, a significant difference in the weight of mock- and BmLV-infected larvae was observed. This pathogenic effect was measured on days 3 and 4 post-infection and followed the peak of viral titers which was on days 1 and 2 post-infection (Figure 5 and Figure 6). Interestingly, this weight difference coincided with the initiation of pathogenesis in IOZCAS-*Ha*-I cells (Figure 2 and Figure 5). In the testes, 6 days post-infection, viral loads reached >100% of those found in Hi5 cells (Figure 6d). This appears particularly high, especially when compared to what has been reported for *B. mori*, where four days after infection the viral load in the infected testes corresponded to 0.08% of that found in *Bm*N4 cells [15]. Surprisingly, the outcome of infection in *H. armigera* was highly variable, with large variation in viral titers between tissues and between replicates from the same treatment condition, even for pooled samples (Figure 6b–d). While the causal mechanisms for this variability have not been established, the main determinant likely related to the physiological properties of individual larvae and tissues. This inference can be made as the same viral inoculate was administered to all animals, and the initial conditions (developmental timing, larval weight, injected viral volume) were highly similar, as shown by the small variation in viral titers at the T0 timepoint (Figure 6). Notably, the virus was never cleared from the tested *H. armigera* pools, as the average viral titer remained above (the high) T0 levels until the adult stage (Figure 6). In fact, the infection timeline was characterized by an acute phase from T0 to day 2, and a persistent phase from day 3 onwards (Figure 6). The switch between phases could be mediated by the activation and functioning of host immune pathways, such as sRNA-mediated pathways, as discussed below. 

A survival curve analysis could not detect a significant difference between mock- and virus-infected animals (Figure 5c). This might have been due to the high number of censoring events for both conditions. While over 90 subjects were at risk per condition on day 0 of observations, this number rapidly decreased as the animals were picked for sampling and dissections. By day 22, less than 20 subjects were at risk per condition (Appendix A). Future experiments with a higher subject number and reduced censoring events could aid in detecting differences in long-term mortality between the experimental conditions.

Notably, by demonstrating that BmLV was capable of infecting *T. ni* and *H. armigera* cells (Figure 2, Figure 3, Figure 4, Figure 5 and Figure 6), we showed that this is not a *B. mori*-specific virus. In fact, *Trichoplusia* and *Helicoverpa* belong to the family of the Noctuidae (superfamily Noctuidea), whereas *Bombyx* belongs to the family Bombycidae (superfamily Bombycoidea), showing that BmLV has a relatively broad species tropism. In addition, since acute symptoms during the infection were observed in IOZCAS-*Ha*-I cells and in *H. armigera* larvae (Figure 2 and Figure 5), we consider that the name BmLV (*Bombyx mori latent virus*) is not a good descriptor. Therefore, we suggest that these newly identified viral characteristics be considered when the species name is adapted to comply with the recently ratified ICTV binomial classification format.

In this paper we also report that BmLV replication colocalized with mitochondria (Figure 3). While the maculaviruses’ site of replication in insect cells has not been reported previously, the type species *Grapevine Fleck Virus* (GFkV) replicates in multivesiculate bodies of plant cells. These are mitochondria with extended vesiculation resulting from the invagination of both lamellae of the boundary membranes [63]. Thus, together with the available literature, our results indicated that BmLV replicates in the mitochondria of insect cells.

### 4.4. sRNA-Based Immune Response to BmLV Infection

In both studied lepidopteran systems, a differential regulation of RNAi pathway genes at the transcriptional level took place. Specifically, regarding the siRNAi pathway, an upregulation of d*cr2* and a*go2* was observed. While this was temporary in the IOZCAS-*Ha*-I cells, it was persistent in *H. armigera* animals (Figure 4a,b and Figure 7a,b). In parallel, it is interesting to notice that while cell mortality increased until the end of the experiment in IOZCAS-*Ha*-I cells; in *H. armigera*, following an initial acute phase of infection with a peak in viral levels, the viral titers decreased and remained stable until adulthood, with relative transcript abundances corresponding to those immediately after injection (Figure 7). Thus, it is interesting to consider that a persistent upregulation of *dcr2* and *ago2* might contribute to the establishment of a persistent BmLV infection in *H. armigera* in vivo. Regarding the piRNA pathway, a downregulation of *siwi* was observed in IOZCAS-*Ha*-I cells from 3 days post infection until (at least) 8 days post-infection (Figure 4c). In *H. armigera*, while an upregulation was observed at 2 days post-infection, this was followed by a decrease to the initial levels (Figure 7c). In this case it is also possible to corelate these transcript levels with the outcome of the viral infection: for instance, it is interesting to hypothesize that an upregulation of *siwi* might be involved in the establishment of the persistent infection in *H. armigera*; while a downregulation corelates with the acute infection observed in IOZCAS-*Ha*-I cells until the end of the observations. In line with this idea, piRNAi has been hypothesized to be involved in the transition from an acute defense mechanism to the establishment of a persistent infection [20]. In addition, PIWI proteins were found to play a role in antiviral immunity of the *Bm*N4 and Hi5 cell lines [2,18], even when viral piRNAs were not detected [18]. Thus, independent of the observed piRNA-sized viral sRNAs discussed below, the observation that PIWI proteins were regulated during infection highlights their potential involvement in the antiviral response in these lepidopteran models. Of note, our measurements were performed at the transcript level and at fixed timepoints, making it impossible to exclude that temporary up- or downregulations were missed. Therefore, together with the fact that the half-life of these proteins remains unknown in the different species and cells, it was not possible to directly correlate a potential role of these proteins with the infection effects. Nevertheless, our observations supported the well-known role of the siRNAi pathway as the main, and most broadly acting, antiviral response in insects [16,17]. In addition, they were in line with the reported antiviral role of PIWI proteins in other lepidopteran cell lines [2,18].

In *H. armigera*, *siwi* genes were found in two distinct genomic loci of two chromosomes, indicating a gene duplication event. By comparing the loci in *H. armigera* to those of the closely related species *T. ni* (Appendix A), it may be that the gene on chromosome 17, which is adjacent to a conserved IS630 family transposase domain, resulted from a transposition event and is a duplicate of the gene on chromosome 12. The high similarity between these paralogs (Appendix A) suggests that the duplication event is evolutionarily recent, and that functional divergence might not have occurred yet. Still, it is unclear how the transposed *siwi* copy is transcriptionally regulated and whether this impacts the functionality of piRNAi in this species and others, warranting further research.

Both in the IOZCAS-*Ha*-I cell line and in the investigated *H. armigera* tissues, sRNAs mapping to BmLV were mainly 20 nt long (Figure 8a), corresponding to the expected size of siRNAs [64]. In fact, viral siRNAs (20 nt, sense and antisense) varied between 53 and 9796 RPM in the different samples of *H. armigera*; and were 3100 RPM in IOZCAS-*Ha*-I cells. PiRNA-sized reads (26–29 nt) mapping to the BmLV genome were identified in both cases. Although they represented between 0.2 and 5% of the viral sRNA reads in *H. armigera* samples, they represented 27% of reads in the IOZCAS-*Ha*-I cell line (Figure 9a). Specifically, viral piRNA-sized reads (26–29 nt, sense and antisense) varied between 1 and 88 RPM in *H. armigera*; and were 1516 RPM in IOZCAS-*Ha*-I cells. To the best of our knowledge, this is the first report of abundant vpiRNA production in a lepidopteran somatic cell line.

In *H. armigera* testes, an siRNA-sized peak of >500 RPM was observed, with a smaller piRNA-sized peak of <10 RPM (Figure 8a and Figure 9a). Of note, in the testes there was a more even balance of sense and antisense piRNA-sized reads compared to other tissues, and the reads peaked at a size of 26 rather than 27 nt. This might indicate differences in the biogenesis of vpiRNAs between the germ line and somatic tissues. While the sRNA response to BmLV infection has not been studied in the testes of other species, it is interesting to note that in the ovaries of *B. mori*, only reduced quantities of 21 and 27 nt sized reads (<1 RPM each) mapping onto the BmLV genome have been shown [15]. 

Another interesting note is that, among the analyzed samples, the IOZCAS-*Ha*-I cells displayed the highest levels of viral piRNA-sized reads while, within the *H. armigera* investigated samples, the highest quantities were represented by the fat body. This is interesting considering that this cell line was derived from final-instar larval fat body [23]. Much lower levels of viral piRNA-sized reads were identified in the fat body of *B. mori* upon infection with the *B. mori* nucleopolyhedrovirus (BmNPV) [65]. Together, these observations highlighted the differences in the BmLV infection process, as well as in the sRNA-based antiviral response, in the different species and/or tissues. 

Regarding the BmLV piRNA-sized reads in IOZCAS-*Ha*-I cells, these were almost exclusively sense oriented and with a weak 1 U bias (Figure 9a and Appendix A), indicating a biogenesis via the primary piRNA pathway. Furthermore, no 9 nt read distance was identifiable between sense and antisense reads, meaning that the ping-pong pathway was not involved (Appendix A). On the other hand, no piRNA fingerprints (nucleotide biases or read distance biases) were found in *H. armigera* for the piRNA-sized reads mapping onto the BmLV genome in any of the tested tissues or developmental stages (Appendix A). This suggests that the relatively few piRNA-sized reads mapping onto BmLV detected during our experiments were not produced through the secondary or tertiary piRNA biogenesis pathways. This finding is in line with observations made in *B. mori* infected with the *B. mori* cypovirus (BmCPV) [66] and BmNPV [65]. Specifically, these studies reported on a relatively low quantity of viral sRNAs in the size range of piRNAs, with no typical piRNA fingerprints, in larval fat body and midgut. 

When looking at the distribution of sRNAs (18–31 nt) over the viral genome, a hotspot was observed at the 5′ end of the BmLV genome in all cases (Figure 8b). However, a clear hotspot at the 3′ end, corresponding to subgenomic RNAs, was only present for the IOZCAS-*Ha*-I cell line (Figure 8b). When focusing on piRNA-sized reads, a hotspot at the 3′ end of the BmLV genome was observed for both the IOZCAS-*Ha*-I cell line and the *H. armigera* fat body, emphasizing the similarities of these two fat-body-derived samples. Overall, while viral sRNAs with the typical size of piRNAs were observed in IOZCAS-*Ha*-I cells and in *H. armigera* (whole larvae and adults; as well as in larval testes, fat body, CNS, and midgut), these were much more prominent in the cell line; and only marginally contributed to the sRNA pool in vivo. Their biogenesis pathway, as well as their role in the antiviral response, remain to be determined in future research. 

### 4.5. Secondary piRNA Biogenesis Pathway in Somatic vs. Germline Cells

In parallel, we identified a fully functional secondary piRNA biogenesis pathway in every analyzed case, specifically in IOZCAS-*Ha*-I cells and in all *H. armigera* samples (Appendix A). While this pathway is often considered germline-specific, there is mounting evidence that in the last common ancestor of arthropods this may have been active in both somatic and germline cells. This is still the case for many insect species, such as the lepidopterans *T. ni* and *B. mori*. Some exceptions are the genera *Drosophila* and *Bombus*, where the somatic function has been lost [67,68,69].

## 5. Conclusions

In this study we described the genomic diversity of BmLV, as well as characterized a BmLV variant derived from the *T. ni* Hi5 cell line. We also demonstrated that BmLV could cause infections in *T. ni*-derived cells, as well as in *H. armigera* in vivo. In fact, the virus had a broader species tropism than initially thought, as it is not a *B. mori*-specific virus. These infections displayed acute symptoms and thus the species name BmLV (*Bombyx mori latent virus*) is not a good descriptor. This should be considered when the species name is adapted to comply with the ICTV binomial classification format. In addition, we demonstrated the regulation of siRNAi and piRNAi proteins during infection, as well as studied the profile of sRNAs mapping to BmLV upon infection in these systems. We showed the abundant production of virus-derived piRNA-sized reads in a lepidopteran somatic cell line. While we detected mainly siRNAs in vivo, some sRNAs in the size range of piRNAs were also noticed. We also discussed the BmLV genomic diversity and the presence of viral co-infections in lepidopteran cell lines. We hope to create awareness of the potential implication of these findings on previous and future research.

## Figures and Tables

**Figure 1 viruses-15-01183-f001:**
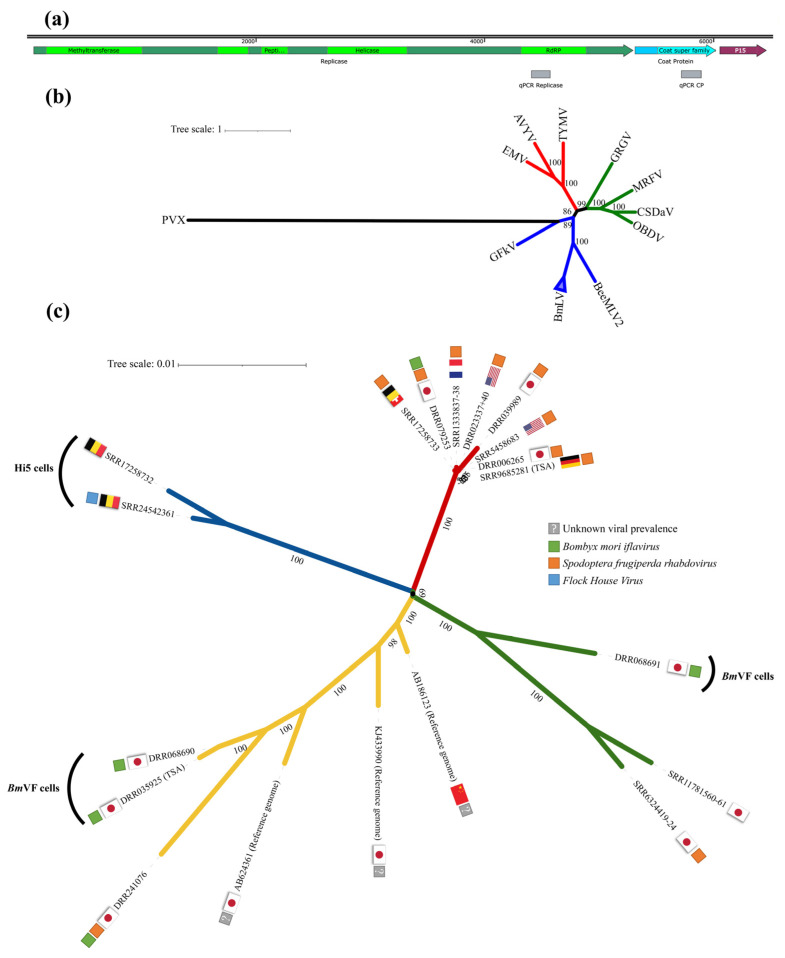
(**a**) Genomic organization of the BmLV. Open reading frames (ORFs) are indicated by arrows. Green: ORF of the replicase polyprotein. Blue: ORF of the capsid protein. Red: ORF of the P15 protein. Conserved domains are indicated by lighter coloration and predicted with the NCBI conserved domain search tool [45]. Grey boxes indicate PCR amplicons used during qPCR analyses. (**b**) Maximum likelihood phylogenetic analysis of BmLV genomes using the translated amino acid sequence of the replicase polyprotein. Red clade: Tymoviruses. Green clade: Marafiviruses. Blue clade: Maculaviruses. The BmLV branch is collapsed and indicated with a triangle, it clustered with another insect maculavirus, namely the BeeMLV2. A Potexvirus was used as the outgroup. Branch identifiers are listed in Appendix A. (**c**) Maximum likelihood phylogenetic analysis of BmLV genomes using the nucleic acid sequence of the replicase polyprotein ORF. The BmLV genomes clustered into four distinct clades, shown by the green, yellow, blue, and red branch colors. Samples are named by the SRA identifiers used to assemble the genomes. Co-infections are indicated by colored boxes; samples are *Bm*N4-derived unless indicated otherwise.

**Figure 2 viruses-15-01183-f002:**
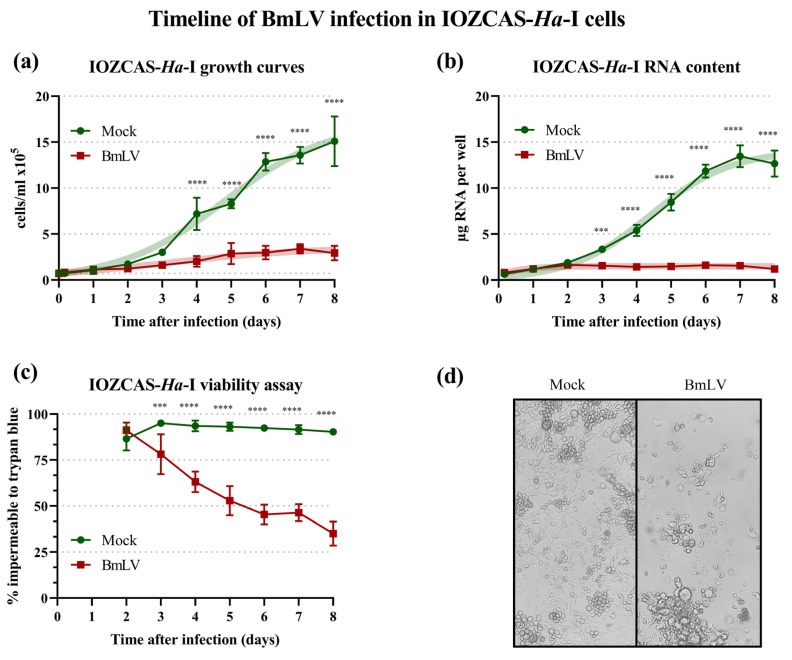
BmLV infection in IOZCAS-*Ha*-I cells. (**a**) Cells treated with BmLV displayed stunted growth compared to mock-infected cells, with significantly different best-fit logistic growth models (*p* < 0.0001). (**b**) Total RNA content of BmLV-infected cells compared to mock-infected cells, with significantly different best-fit logistic growth models (*p* < 0.0001). (**c**) Infection with BmLV was associated with high mortality compared to mock-infected cells. Cell viability was assayed with trypan blue staining. (**d**) Micrograph (100× amplification) of mock- and BmLV-infected cells on day 7 after treatment. Symbols indicate the mean, and error bars the standard deviation. Four replicates were tested per condition. Asterisks indicate a significant difference between mock- and BmLV-infected samples from the same timepoint, as determined using a two-way ANOVA with post hoc Sidak’s multiple comparisons test (*** *p* ≤ 0.001; **** *p* ≤ 0.0001). Thick shaded lines in graphs (**a**,**b**) show the best-fit logistic growth model for each condition.

**Figure 3 viruses-15-01183-f003:**
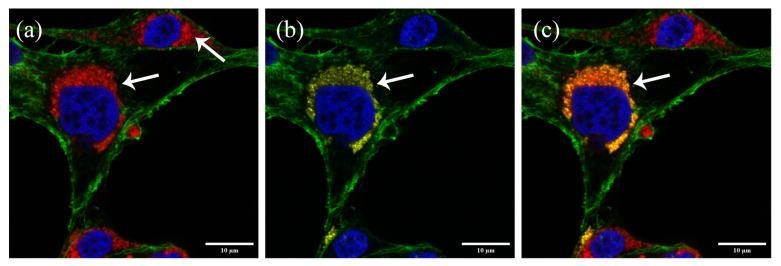
BmLV associates with mitochondria during replication. IOZCAS-*Ha*-I cells were infected with BmLV and imaged 1 day after with confocal immunocytochemistry. (**a**) MTDR stained mitochondria in cells with and without viral replication (lower and upper arrow, respectively). (**b**) dsRNA accumulated in the sites of viral replication. (**c**) Overlaying the MTDR and J2 signals suggested colocalization of viral replication sites with mitochondria. Blue: DAPI, red: MTDR, green: Phalloidin–TRITC, yellow: J2–Alexa Fluor™ 488.

**Figure 4 viruses-15-01183-f004:**
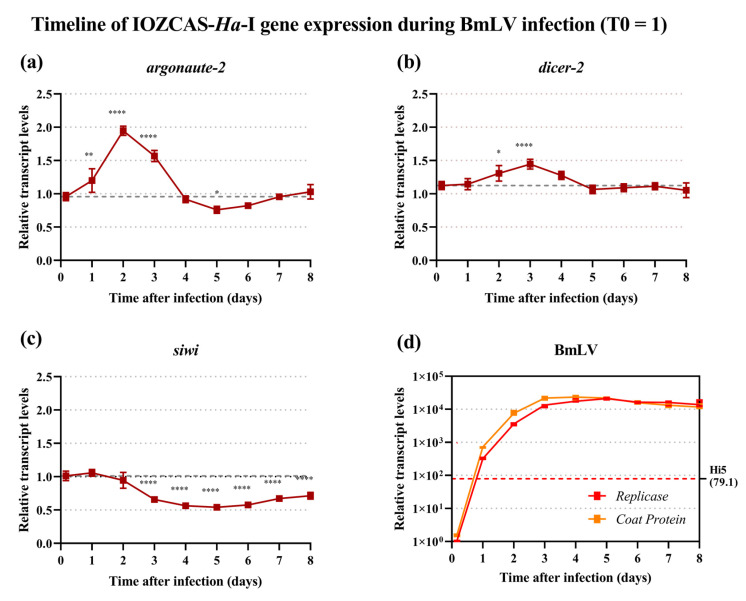
Relative transcript levels in IOZCAS-*Ha*-I cells of selected RNAi genes and BmLV during the progression of infection. The transcript levels of siRNA-pathway genes (**a**) *ago2* and (**b**) *dcr2*; of the piRNA-pathway gene (**c**) *siwi*; and of (**d**) the BmLV titers were assayed at multiple timepoints via RT-qPCR. Relative values were obtained via the ddCt method normalizing towards the T0 timepoint. Four replicates per condition. Symbols indicate the mean, error bars the standard deviation. The grey dotted line (**a**–**c**) indicates the mean transcript level at the 4 h timepoint. The red dotted line (**d**) indicates the relative BmLV *rep* levels in Hi5 cells. For figures (**a**–**c**), the time-dependent variation in transcript levels was assayed via an ordinary one-way ANOVA with post hoc Dunnett’s multiple comparison test towards the 4 h timepoint. Asterisks indicate a significant difference in the mean transcript levels towards the 4 h timepoint (* *p* ≤ 0.05; ** *p* ≤ 0.01; **** *p* ≤ 0.0001).

**Figure 5 viruses-15-01183-f005:**
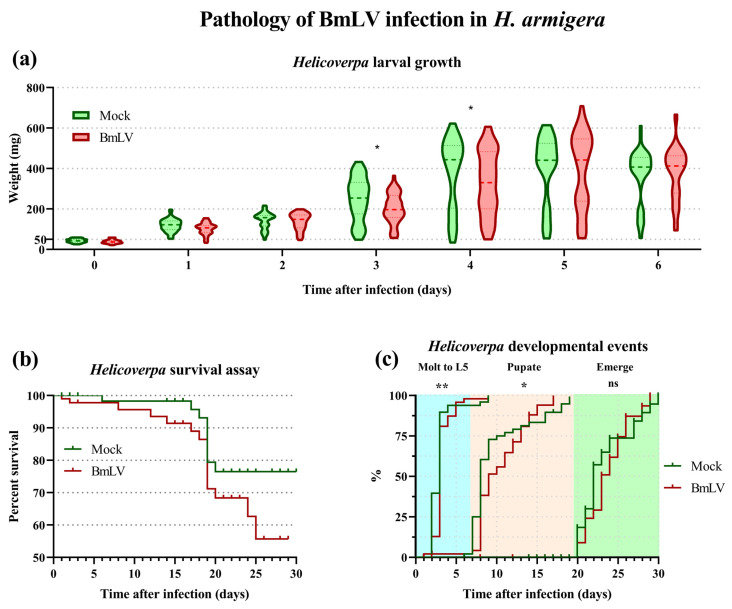
Pathology of BmLV infection in *H. armigera*. Early L4 larvae weighing 60 mg or less were injected with 3 µL of control or BmLV-infected cell extract. (**a**) Violin plot comparing the weight of mock- or virus-treated larvae for a six-day period. The weights differed significantly at day 3 and 4 after infection (multiple *t*-tests, 48 animals per condition). Thick dotted lines indicate the median, thin dotted lines the quartiles. (**b**) Survival assay of mock- or BmLV-infected *H. armigera*. No significant difference was found through a Mantel–Cox or Gehan–Breslow–Wilcoxon test (*p* > 0.14). Ticks indicate censoring events. Subject numbers provided in Appendix A. (**c**) Infection with BmLV significantly affected development of *H. armigera* (Gehan–Breslow–Wilcoxon test, 48 animals per condition). * *p* ≤ 0.05; ** *p* ≤ 0.01.

**Figure 6 viruses-15-01183-f006:**
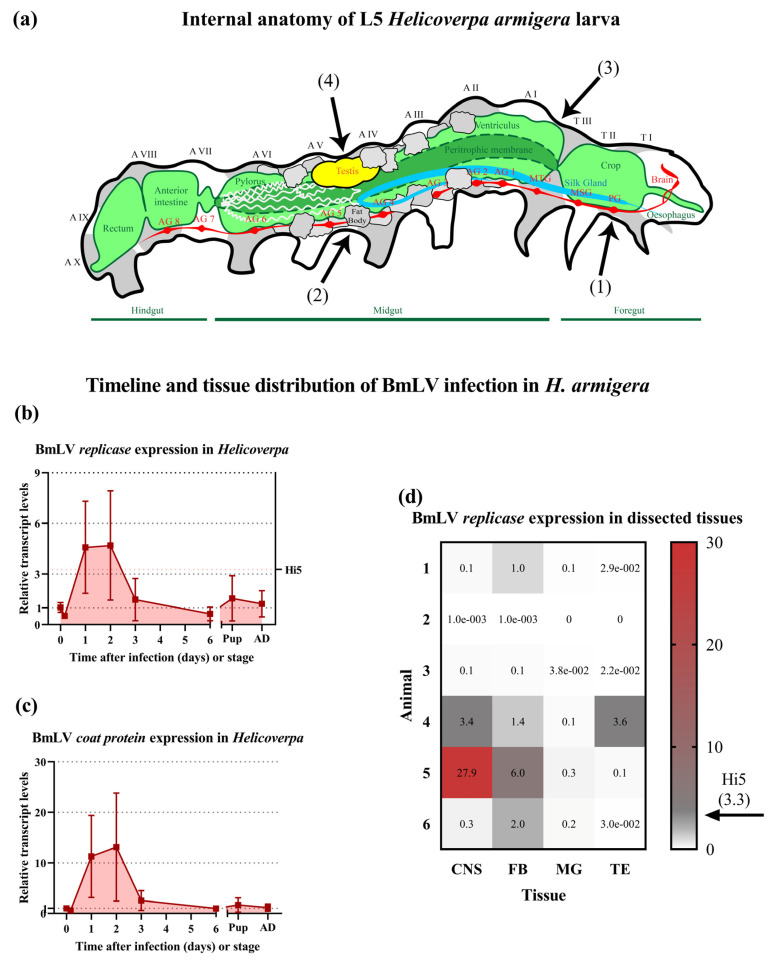
(**a**) Schematic diagram of the internal anatomy of an L5 *H. armigera* larva, with dissected tissues. (**a1**) The central nervous system is composed of the brain (or supra-esophageal ganglia) and the segmental ganglia of the ventral nerve cord. (**a2**) The fat body consists of thin sheets of tissue dispersed throughout the body cavity (for simplicity in this scheme it is displayed only in segments A II until A VI), located under the integument and surrounding the gut and testes. (**a3**) The midgut is the largest internal organ and occupies most of the abdominal segments. (**a4**) The pair of testes are bean-shaped and located dorsally at the fifth abdominal segment. (**b**–**d**) Timeline and tissue distribution of BmLV infection in *H. armigera*. Transcript levels of the BmLV *replicase* (**b**,**d**) and *coat protein* (**c**) genes were assayed at multiple timepoints (**b**,**c**) or in day 6 dissected tissues (**d**) via RT-qPCR. Expression levels were normalized towards the T0 timepoint using the ddCt method. For the timeline plots (**b**,**c**) symbols indicate the mean, error bars the standard deviation. Pup = pupal stage, AD = adult stage. Four replicates per timepoint. For the day 6 tissue distribution heatmap (**d**), rows indicate individual larvae and columns the dissected tissue. CNS = central nervous system, FB = fat body, MG = midgut, TE = testes. The BmLV *replicase* titer in Hi5 cells was similarly compared to *H. armigera* T0. The BmLV *rep* expression value in Hi5 cells is indicated by the red dotted line (**b**) or the black arrow (**d**).

**Figure 7 viruses-15-01183-f007:**
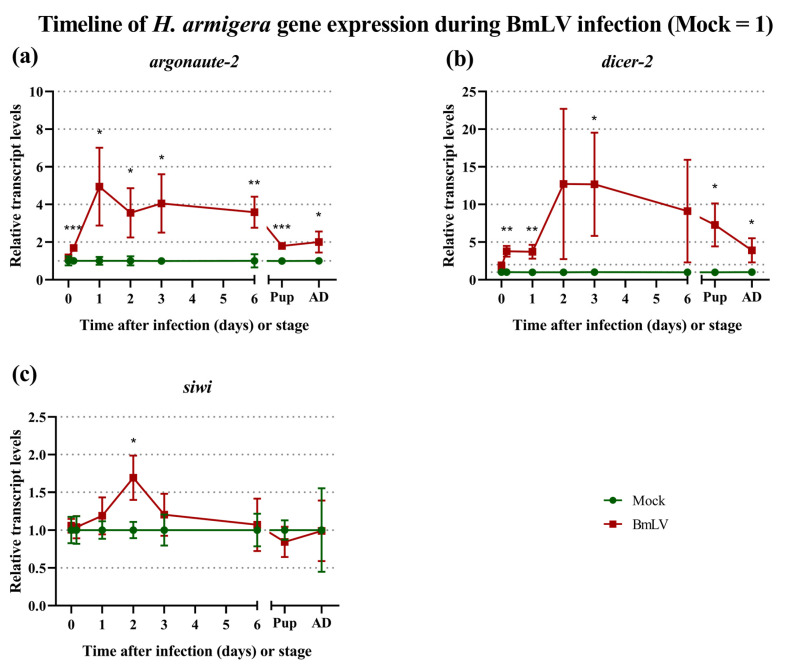
Timeline of *H. armigera* RNAi gene expression during BmLV infection. The transcript levels of three RNAi immune pathway genes, (**a**) *ago2*, (**b**) *dcr2*, and (**c**) *siwi*, were monitored from the moment of infection in L4 larvae (T0) until the adult stage via RT-qPCR. The transcript levels were normalized via the ddCt method towards the mean of the control (mock-infected) samples of the same timepoint. Symbols indicate the mean, error bars the standard deviation. Four pools of three entire animals were tested per condition. Asterisks indicate a significant difference in the mean values between the mock- and BmLV-infected samples from the same timepoint, as determined via multiple t-tests (* *p* ≤ 0.05; ** *p* ≤ 0.01; *** *p* ≤ 0.001).

**Figure 8 viruses-15-01183-f008:**
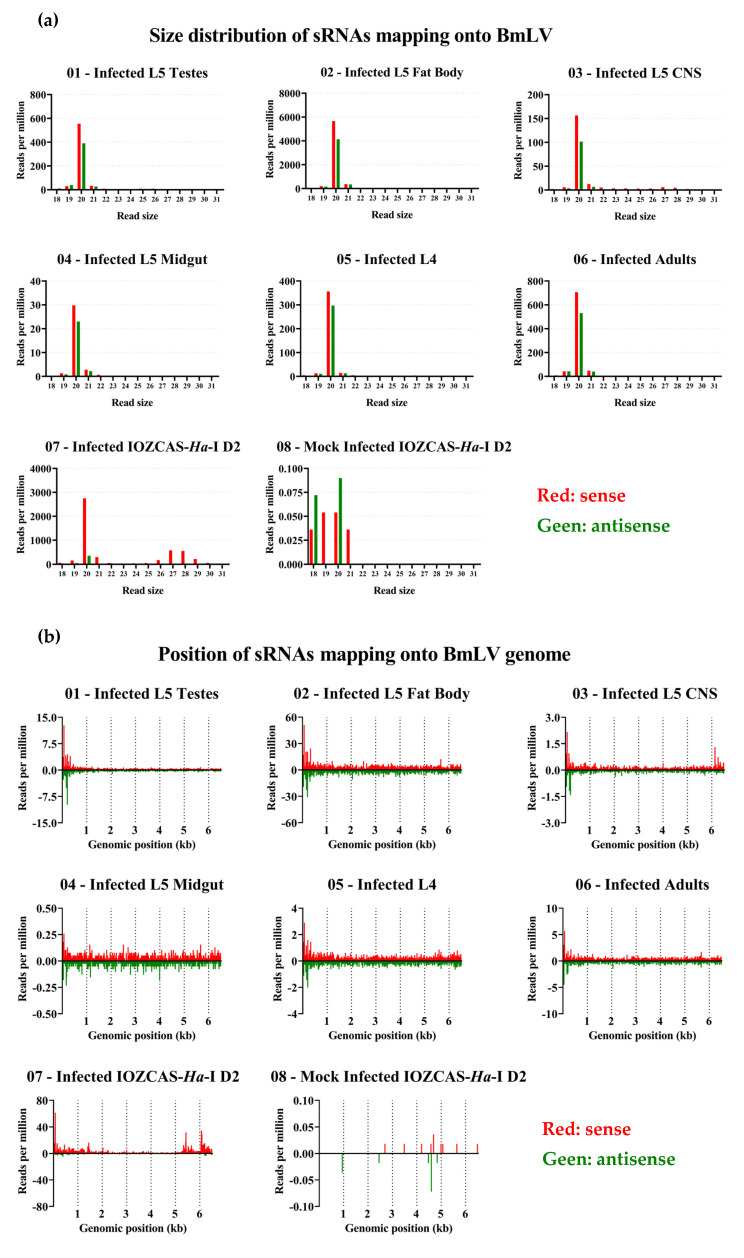
Analysis of sRNAs mapping onto the BmLV genome sampled from *H. armigera* animals (samples 01 to 06) and IOZCAS-*Ha*-I cells (samples 07 and 08). Tissues were dissected on day 6 after infection, whole larvae were collected on day 2 after infection, whole adults were collected on day 1 of the adult stage, and the cells were collected on day 2 after (mock) infection. (**a**) Length distribution of sRNAs mapping onto the BmLV genome expressed in reads per million (from 18 to 31 nt). (**b**) Positional frequency distribution of sRNAs mapping onto the BmLV genome expressed in reads per million.

**Figure 9 viruses-15-01183-f009:**
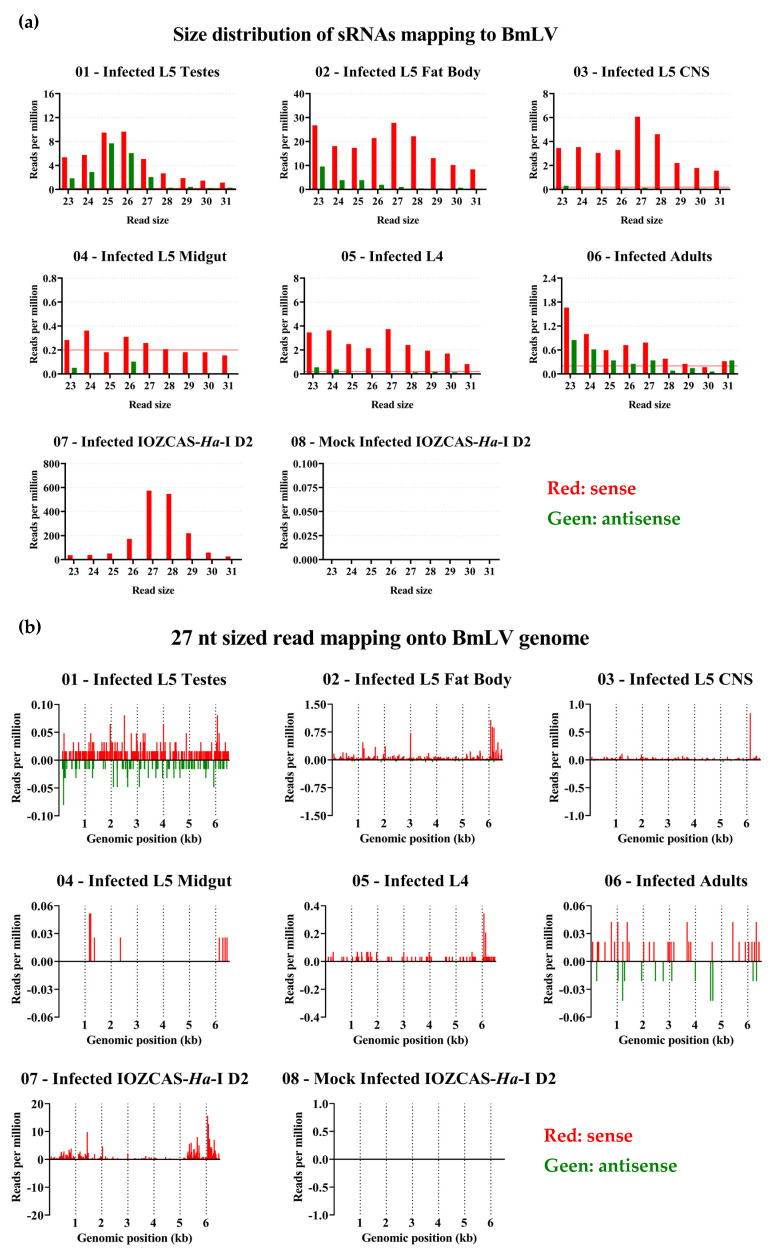
Analysis of piRNA-sized sRNAs mapping onto the BmLV genome sampled from *H. armigera* animals (samples 01 to 06) and IOZCAS-*Ha*-I cells (samples 07 and 08). Tissues were dissected on day 6 after infection, whole larvae were collected on day 2 after infection, whole adults were collected on day 1 of the adult stage, and the cells were collected on day 2 after (mock) infection. (**a**) Frequency of sRNAs mapping onto the BmLV genome expressed in reads per million and divided by the read size (from 23 to 31 nt). (**b**) Positional frequency distribution of 27 nt sized sRNAs mapping onto the BmLV genome expressed in reads per million.

## Data Availability

The *Helicoverpa armigera* and IOZCAS-*Ha*-I sRNA-Seq datasets obtained in this study were deposited on NCBI under BioProject PRJNA972300.

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
