# Peer review of "Identification and Profiling of a Novel Bombyx mori latent virus Variant Acutely Infecting Helicoverpa armigera and Trichoplusia ni"

_viruses, 2023, doi:10.3390/v15051183_

Round 1

Reviewer 1 Report

Foreword I would like to point out that the only PDF to which I had access did not contain all the supplementary data.

In this manuscript, Verdonckt et al. collected several SRA libraries to identify and assemble BmLV genomes. BmLV are found in all the BmN4 cells bioprojects. Including their own strain infecting Hi5 cells, they aligned 16 genomes to build a tree with maximum likelihood. Subsequently, they focused on a strain by infecting IOZCAS-Ha-I cells and then by infecting lepidopteran H. armigera larvae. They demonstrated that this strain was capable of infecting these cells as well as the larvae by characterizing the siRNA and piRNA response to this infection but without noting a significant mortality compared to the controls. If the siRNA response seems very comparable between the different tissues tested, they showed a more tissue specific piRNA response.

The manuscript is well written, the methodology and the results are clearly exposed.

Major comment :

Figure 1 : A phylogenetic tree must always be presented with its bootstrap results otherwise they are useless.

Minor comment :

line 155 : As the reference is only related to the Ugene software and not to the method, the authors should better explain their methodology to detect true SNPs.

line 205 : “prepared in paragraph 0” ?

line 460 : Multiple alignments should be provided as supplementary data.

line 461 : To get insight into the evolutionary history of the 16 genomes, I do not understand why the authors did not use the entire genomes. The intergenic parts can be richer in information allowing to classify these strains.

Line 736 : The size distribution of all small RNAs from each library should be provided as additional data. A significant variation in the amount of other small RNAs (miRNAs, anti transposon piRNAs) has a strong impact on the amount of RNAs mapping to the virus.

Author Response

The remarks of the reviewer are written in bold. The response of the authors is written in regular script:

Foreword I would like to point out that the only PDF to which I had access did not contain all the supplementary data.

This is very unexpected. We had uploaded a pdf of the manuscript, a pdf of the supplementary data and a zip-archive with the manuscript and supplementary word-documents. We are not aware why reviewer 1 did not have access to the other files.

In this manuscript, Verdonckt et al. collected several SRA libraries to identify and assemble BmLV genomes. BmLV are found in all the BmN4 cells bioprojects. Including their own strain infecting Hi5 cells, they aligned 16 genomes to build a tree with maximum likelihood. Subsequently, they focused on a strain by infecting IOZCAS-Ha-I cells and then by infecting lepidopteran H. armigera larvae. They demonstrated that this strain was capable of infecting these cells as well as the larvae by characterizing the siRNA and piRNA response to this infection but without noting a significant mortality compared to the controls. If the siRNA response seems very comparable between the different tissues tested, they showed a more tissue specific piRNA response.

The manuscript is well written, the methodology and the results are clearly exposed.

 Major comment:

  • Figure 1: A phylogenetic tree must always be presented with its bootstrap results otherwise they are useless.

Thank you for this important observation. We have added the bootstrap values to the phylogenetic trees.

Minor comments:

  • line 155: As the reference is only related to the Ugene software and not to the method, the authors should better explain their methodology to detect true SNPs.

 We have added a description of our method to section 2.1.2 of materials and methods.

  • line 205: “prepared in paragraph 0” ?

This mistake originated from a broken cross-reference in the text, resulting in the wrong paragraph number. We have corrected this issue.

  • line 460: Multiple alignments should be provided as supplementary data.

We have provided an MView generated multiple alignment of the replicase amino acid sequences in the supplementary data (Supplementary text 1).

  • line 461: To get insight into the evolutionary history of the 16 genomes, I do not understand why the authors did not use the entire genomes. The intergenic parts can be richer in information allowing to classify these strains.

This is a very good observation. We performed the phylogenetic analysis using the replicase polyprotein amino acid sequences as one of our goals was to assess the relationship of BmLV with other viruses from the Tymoviridae family. As between genera the viral genomes are very different, both in terms of organization and sequence, we found that using the entire genomic sequence was not optimal for performing this phylogenetic analysis. On the other hand, the replicase polyprotein amino acid sequences were sufficiently conserved and resulted in an accurate phylogenetic tree (see added bootstrap values in revised manuscript). The results can be seen in Figure 1B.  After considering your feedback, we realized that the replicase polyprotein amino acid sequences might however not be ideal for determining the relationship between BmLV variants, as some information is lost during the translation process. For this reason, we repeated the analysis using the nucleic acid sequences of the replicase polyprotein ORFs. This resulted in an accurate phylogenetic tree (see the bootstrap values of Figure 1C in the revised manuscript). However, the relationship with other Tymoviridae was no longer determined and the tree is unrooted. We again only used the polyprotein sequence as the BmLV genome has very few intergenic parts and the coat protein and CP sequences are more conserved.

  • Line 736: The size distribution of all small RNAs from each library should be provided as additional data. A significant variation in the amount of other small RNAs (miRNAs, anti transposon piRNAs) has a strong impact on the amount of RNAs mapping to the virus.

We had already prepared a graph showing the size distribution of the reads for each sample. We did not initially add it to the manuscript as we did not find it presented important information. However, with the reasoning provided by reviewer 1, it makes sense that it should be added to the supplementary information. It is now Supplementary Figure 4.

Reviewer 2 Report

Verdonckt et al. present the results of a detailed study on BmLV in a lepidopteran cell line and lepidopteran larvae (H. armigera). They observed marked pathogenic characteristics in both systems following inoculation and virus tissue tropism was characterized and shown to extend to gonadal tissue. The RNAi response was characterized in detail and PIWI proteins were shown to be regulated during infection. Virus-derived piRNAs were particularly abundant in cultured cells. The study was well performed, is original and represents a detailed and valuable contribution to the understanding of the host-virus relationship in vitro and in vivo systems.
I have only a few minor issues that the authors may wish to address (see scanned copy of the manuscript attached):
Numbered points:
1. Suggest you include the name of the virus in the title (not just the abbreviation).
2. Virus names are not usually revised just because some new characteristic has been recognized; suggest they look at the ICTV website to determine under which circumstances a name change would be considered.
3. Genus names are italicized.
4. If referring to the common name, virus names are not italicized even when they contain the latinized name of the host. If referring to the species name (BmLV is a recognized species), the entire name is italicized.
5. Insert "isolated from insects".
6. Please spell out words of each abbreviation at first use (entire manuscript). This manuscript uses a lot of abbreviations.
7. I did not understand the concept of "paragraph 0". Please revise.
8. Where did the insect colony originate from? How long in culture?
9. Ethanol treatment would result in surface decontamination, not sterilization.
8. (error in my numbering) line 424: if the data did not have equal variance did you employ Welch's adjustment for t-test comparisons, or how did you handle this?
9. (line 570): are the ± values SEs or SDs?
10. Survival curves look different but are stated to be non-significant (P=0.14; Fig 5B): Can you suggest a reason? I note that there was a lot of censoring. Could you explain this briefly?
11. Loaves? This is not the correct term. Maybe thin "sheets" of tissue?
12. I would suggest "appearance of cell mortality (or CPE)" rather than emergence of pathogenicity, as pathogenicity is a virus trait, whereas you are referring to a host response.
13. See point 2 above.
14. Reword to "highlights their potential involvement in the antiviral response...."
13. (misnumbered by me; line 1020): see point 2 above.
Finally, I would ask the authors to present their text in the past tense throughout the manuscript (at present it comprises a mixture of tenses).

Uniformize text to the past tense please. See suggestions written on scanned manuscript.

Author Response

The remarks of the reviewer are written in bold. The response of the authors is written in regular script:

Verdonckt et al. present the results of a detailed study on BmLV in a lepidopteran cell line and lepidopteran larvae (H. armigera). They observed marked pathogenic characteristics in both systems following inoculation and virus tissue tropism was characterized and shown to extend to gonadal tissue. The RNAi response was characterized in detail and PIWI proteins were shown to be regulated during infection. Virus-derived piRNAs were particularly abundant in cultured cells. The study was well performed, is original and represents a detailed and valuable contribution to the understanding of the host-virus relationship in vitro and in vivo systems.

I have only a few minor issues that the authors may wish to address (see scanned copy of the manuscript attached):

Numbered points:

  1. Suggest you include the name of the virus in the title (not just the abbreviation).

We have included the name of the virus in full in the title.

  1. Virus names are not usually revised just because some new characteristic has been recognized; suggest they look at the ICTV website to determine under which circumstances a name change would be considered.

The reviewer is correct that virus names are not adapted whenever new properties are discovered. In line with this, we have changed the text of the manuscript and no longer propose that the name be adapted due to the findings of our manuscript. However, ICTV recently ratified a binomial classification format to which all virus names must comply. Therefore, the BmLV species name will need to be changed and we suggest that these newly identified viral characteristics be considered on that occasion.

  1. Genus names are italicized.

We have italicized all genus names.

  1. If referring to the common name, virus names are not italicized even when they contain the latinized name of the host. If referring to the species name (BmLV is a recognized species), the entire name is italicized.

We have revised the formatting of virus and virus species names throughout the text following the recommendations of the ICTV (as stated on September 16, 2020 - https://ictv.global/filebrowser/download/440).

  1. Insert "isolated from insects".

We think that this suggestion is not entirely applicable. While it is true that the virus has not been isolated from insects, it has also never been detected through analytical methods like PCR, immunoassays, or vdSAR. Therefore, it is more correct and informative to state that the virus has never been detected (rather than isolated).

  1. Please spell out words of each abbreviation at first use (entire manuscript). This manuscript uses a lot of abbreviations.

We have taken care to spell out words at first use. However, on line 448 of the revised manuscript we did not spell-out SRA (as suggested by reviewer 2) as it was already defined on line 123. Likewise, we did not spell out TSA on line 457 as it was already spelled out on line 372.

  1. I did not understand the concept of "paragraph 0". Please revise.

This mistake resulted from a defective cross-link in the original word document. We have corrected the reference.

  1. Where did the insect colony originate from? How long in culture?

We have added the requested information to the manuscript.

  1. Ethanol treatment would result in surface decontamination, not sterilization.

That is correct, we have adapted the text accordingly.

  1. (error in my numbering) line 424: if the data did not have equal variance did you employ Welch's adjustment for t-test comparisons, or how did you handle this?

We have conducted Welch’s t-tests and now specified so in the manuscript.

  1. (line 570): are the ± values SEs or SDs?

We have modified the text to clarify that the values given are standard deviations.

  1. Survival curves look different but are stated to be non-significant (P=0.14; Fig 5B): Can you suggest a reason? I note that there was a lot of censoring. Could you explain this briefly?

The survival curve experiment was performed with all treated larvae, including those that were sampled for the timeline and tissue distribution analyses. This meant that the subject number gradually decreased lowering the power of the analysis. We have added a paragraph in the discussion to clarify this.

  1. Loaves? This is not the correct term. Maybe thin "sheets" of tissue?

Sheets is indeed the correct term, we have adapted the manuscript.

  1. I would suggest "appearance of cell mortality (or CPE)" rather than emergence of pathogenicity, as pathogenicity is a virus trait, whereas you are referring to a host response.

Thank you for this suggestion, we have adapted the text accordingly.

  1. See point 2 above.

See our response to point 2.

  1. Reword to "highlights their potential involvement in the antiviral response...."

We appreciate the suggestion and have applied it in the text.

  1. (misnumbered by me; line 1020): see point 2 above.

See our response to point 2.

Finally, I would ask the authors to present their text in the past tense throughout the manuscript (at present it comprises a mixture of tenses).

We have taken care to adapt the tenses throughout the manuscript in accordance with effective writing guidelines.

Response from the authors:

We highly appreciate the thorough feedback that we received from Reviewer 2 and agree with most points. We have adapted the manuscript text as suggested, but only have minor remarks about some suggestions:

  • On line 428 of the revised manuscript, reviewer 2 advised to change “animals” to “larvae”. However, the experiment described in that sentence was conducted on animals from all developmental stages, including pupae and adults. Therefore, we did not apply the suggestion.
  • On line 794 of the revised manuscript, the reviewer advised to change “the BmLV has not yet been detected in animals” to the past tense “the BmLV had not yet been detected in animals”. However, this statement is true at the time of writing, and changing to the past tense may give the wrong impression that it is no longer true.
  • On line 908 of the revised manuscript, reviewer 2 advised to remove information about the superfamily of the genus Bombyx. We think that it is useful to keep this information as it serves to underline that Bombyx differs from the genera Helicoverpa and Trichoplusia up to the superfamily rank.